# A Generalized Bayesian Approach to Distribution-on-Distribution Regression

**Tin Lok James Ng**[1]

[1]School of Computer Science and Statistics, Trinity College Dublin, Dublin, Ireland

## Abstract

In recent years, there has been growing interest in distribution-on-distribution regression, a regression problem where both covariates and responses are represented as probability distributions. Despite various methodologies proposed to address this challenge, a notable absence has been a Bayesian approach, which offers benefits by allowing for the integration of prior knowledge and providing a formal means of quantifying uncertainty. However, a major challenge in employing a Bayesian approach lies in the complexity of fully specifying the data generating process. To overcome this obstacle, we adopt a generalized Bayesian approach and investigate the contraction rates of the resulting generalized (Gibbs) posterior distributions. We propose an MCMC algorithm to sample from the generalized posterior distribution and conduct simulation studies to validate the theoretical findings. Finally, we apply the model to a data application involving mortality data.

## 1 INTRODUCTION

Performing inference on a set of probability distributions has become increasingly popular in both the statistics and machine learning communities [Poczos et al., 2013, Hron et al., 2016, Petersen and Müller, 2019, Pegoraro and Beraha, 2022, Chen et al., 2023]. However, the inherent non-linear structure of the space of probability distributions poses challenges when applying methods designed for Euclidean or functional data. An approach to address this challenge involves applying a suitable transformation to map the probability distributions to a space with a linear structure [Kneip and Utikal, 2001, Petersen and Müller, 2016]. However, this approach does not consider the geometry of the space of probability distributions, resulting in a non-isometric transformation altering distances between pairs of distributions.

The Wasserstein metric has gained prominence as a tool for measuring the distance between probability distributions. This metric has been applied in various contexts, such as principal component analysis (PCA) of probability distributions, and K-means clustering of probability distributions [Zhuang et al., 2022]. The Wasserstein metric has also found applications in distribution-on-distribution regression, a type of regression modeling where both predictors and responses are distributions. For example, Chen et al. [2023] and [Zhang et al., 2022] use the tangent structure of Wasserstein space to develop linear regression models between tangent spaces. On the other hand, Ghodrati and Panaretos [2022] proposed a regression model using a monotone optimal transport map. Recently, there has been growing research aimed at addressing the problem in higher dimensions Okano and Imaizumi [2023], Ghodrati and Panaretos [2023]. This has proven to be a challenging task due to the lack of closed-form solution for computing the Wasserstein distance in general cases and the curse of dimensionality.

Despite the increasing array of methodologies addressing distribution-on-distribution regression, a Bayesian approach for this problem is notably absent. A Bayesian perspective offers advantages by providing a principled means to incorporate prior information and a formal framework for understanding and quantifying uncertainty associated with the regression operator. In this work, we address this gap by proposing a (generalized) Bayesian framework for distribution-on-distribution regression. Given the potential challenges of fully specifying the data generation process in a standard Bayesian approach, we navigate this issue by adopting the generalized Bayesian framework [Bissiri et al., 2016, Syring and Martin, 2023], which replaces the (negative) log-likelihood function with a loss function.

In this work, our focus is on scenarios where both the covariate and response distributions are defined on the real line. Following the approach in Ghodrati and Panaretos [2022], we directly model the regression operator using a monotone transport map. We parameterize the monotone transport map through Bernstein polynomial basis functions. A natural loss function is introduced, enabling the use of a generalized Bayesian framework for inferring the monotone transport map. We investigate the contraction rates of the (generalized) posterior distribution in two settings: one where the covariate and response measures are directly observed, and another where only consistent estimates of the measures are available, obtained from random samples of each respective measure.

The rest of the article follows this structure: In Section 2, we provide an overview of Wasserstein space, Gibbs posterior distributions, and Bernstein polynomials. We then outline our modeling assumptions for distribution-on-distribution regression, discuss prior specification, examine the contraction rates of Gibbs posterior distributions, and elaborate on the MCMC sampling process in Section 3. Section 4 delves into the details of the simulation studies, while Section 5 focuses on a data application. In Section 6, we explore potential extensions of the current work.

## 2 BACKGROUND

### 2.1 WASSERSTEIN SPACE

Let $\Omega \subset \mathbb{R}$ be a compact interval, let $\mathcal{W}_2(\Omega)$ be the set of probability measures on $\Omega$ with finite second moment. The 2-Wasserstein distance on $\mathcal{W}_2(\Omega)$ is defined as

$$d_{\mathcal{W}}^2(\mu, \nu) = \inf_{\gamma \in \Gamma(\mu,\nu)} \int_{\Omega \times \Omega} |x - y|^2 d\gamma(x, y), \quad (1)$$

where $\mu, \nu \in \mathcal{W}_2(\Omega)$ and $\Gamma(\mu, \nu)$ is the set of couplings of $\mu$ and $\nu$, that is, the set of probability measures on $\Omega \times \Omega$ with marginals $\mu$ and $\nu$. The $\gamma$ that achieves the infinum on the RHS of (1) is said to be the optimal transport plan between $\mu$ and $\nu$.

A deterministic map $T : \Omega \rightarrow \Omega$ is said to be a transport map from $\mu$ to $\nu$ if $\nu = T \# \mu$, that is, $\nu$ is the pushforward measure of $\mu$ under the map $T$, i.e. $\nu(B) = \mu(T^{-1}(B))$ for all Borel sets $B$. Suppose for some $T$, the joint distribution of $(X, T(X))$ achieves the infinum on the RHS of (1), $T$ is said to be an optimal transport map.

When $\mu$ is absolutely continuous with respect to the Lebesgue measure on $\Omega$, the optimal map is given by $T = F_\nu^{-1} \circ F_\mu$, where $F_\mu$ and $F_\nu$ are the cumulative distribution functions for $\mu$ and $\nu$, respectively [Ambrosio et al., 2008, Chapter 6]. In this case, the 2-Wasserstein

distance (1) reduces to

$$d_{\mathcal{W}}^2(\mu, \nu) = \int_0^1 |F_\mu^{-1}(p) - F_\nu^{-1}(p)|^2 dp. \quad (2)$$

### 2.2 GIBBS POSTERIOR DISTRIBUTION

The generalized (Gibbs) posterior distribution [Bissiri et al., 2016, Syring and Martin, 2023] generalizes the standard posterior distribution in Bayesian inference setting by replacing the log-likelihood function with a (negative) loss function. Consider random elements $U_1, \ldots, U_n$ generated from some distribution $P$, and suppose we wish to make inference on a relevant feature of $P$ defined as some functional $\theta = \theta(P)$, taking values in $\Theta$. The Gibbs posterior framework requires specifying a loss function $\ell_\theta(u)$ that measures how closely $\theta$ agrees with a data point $u$. The risk function corresponding to the loss $\ell_\theta(u)$ is

$$R(\theta) = P\ell_\theta(U). \quad (3)$$

Here, $Pf$ denotes the expectation of $f(U)$ with respect to $U \sim P$.

The risk function is unattainable since it depends on the unknown distribution $P$, and thus the inference is conducted using the empirical risk:

$$R_n(\theta) = \frac{1}{n} \sum_{i=1}^n \ell_\theta(U_i).$$

Given a prior distribution $\Pi_\Theta$ on $\Theta$, the Gibbs posterior distribution $\Pi_\Theta^{(n)}$ is defined as

$$\Pi_\Theta^{(n)}(d\theta) \propto e^{-\omega n R_n(\theta)} \Pi_\Theta(d\theta), \quad \theta \in \Theta, \quad (4)$$

where $\omega > 0$ is the so-called learning rate parameter specified by the data analyst. The right-hand side of (4) is assumed to be integrable in $\theta$, and thus the proportionality constant is well defined.

Let $\theta_0$ denote the parameter that minimizes the expected risk (3). Given a semi-metric $d$ on $\Theta$, the Gibbs posterior distribution is said to asymptotically concentrate around $\theta_0$ at rate (at least) $\epsilon_n$, with respect to $d$, if

$$P^n \Pi_n(\{\theta : d(\theta, \theta_0) > M_n \epsilon_n\}) \rightarrow 0 \quad (5)$$

as $n \rightarrow \infty$, where $M_n \rightarrow \infty$ arbitrarily slowly or $M_n = M$ for some large constant $M$. Here $P^n f$ denotes the expectation of $f(U_1, \ldots, U_n)$ where $U_1, \ldots, U_n$ are an i.i.d. sample from $P$. The general conditions for contraction of Gibbs posterior distribution were studied in Syring and Martin [2023].

## 2.3 BERNSTEIN POLYNOMIALS

Bernstein polynomials (BP) basis functions are a popular choice for monotone and shape-constrained regression [Chak et al., 2005, Curtis and Ghosh, 2011, Wilson et al., 2020]. The $k$th BP basis function of order $K$ is

$$b_K(x,k) = \binom{K}{k} x^k (1-x)^{M-k},$$

for $k = 0, 1, \ldots, K$ and for $x \in [0, 1]$. An unknown regression function $f : [0, 1] \to \mathbb{R}$ can then be modelled as

$$f(x) = \sum_{k=0}^{K} \beta_k b_K(x,k). \qquad (6)$$

Requiring $f$ to be monotonically increasing is equivalent to requiring $\beta_k \leq \beta_{k+1}$, $k = 0, \ldots, K-1$. Curtis and Ghosh [2011] uses the re-parameterization $\theta_0 = \beta_0$, and $\theta_k = \beta_k - \beta_{k-1}$ for $k = 1, \ldots, K$. It can be shown that (6) can then be written as

$$f(x) = \theta_0 + \sum_{k=1}^{K} \theta_k G_{B(k,K-k+1)}(x), \qquad (7)$$

where $G_{B(k,K-k+1)}$ is the cumulative distribution function of the Beta distribution with parameters $k$ and $K-k+1$. This re-parameterization is advantageous when the target monotone function exhibits relative flatness within certain sub-intervals $(a, b) \subset [0, 1]$ [Curtis and Ghosh, 2011]. For finite $K$, the BP basis functions with these constraints do not span the entire class of continuous monotonic functions. However, as $K \to \infty$, any continuous monotonic function $f : [0, 1] \to \infty$ can be increasingly well approximated using BP basis functions [Chang et al., 2007].

## 2.4 OTHER RELATED WORKS

There are other regression problems related to the distribution-on-distribution regression problem, such as real-valued responses paired with distribution predictors [Law et al., 2018], distributional responses paired with Euclidean predictors [Han et al., 2020], and regression with manifold-valued data [Shi et al., 2009, Lin et al., 2017]. A distinctive feature of the distribution-on-distribution regression problem is that it typically involves working with Wasserstein space and optimal transport theory.

## 2.5 ADDITIONAL NOTATIONS

We use $a \lesssim b$ to denote that there exists a constant $C$ such that $a \leq Cb$ holds, and $a \asymp b$ to denote that $a \lesssim b$ and $b \lesssim a$. Given a measure $\mu$, the $L^p$ norm of a function $f$ is denoted by $||f||_{L^p(\mu)}$.

# 3 DISTRIBUTION-ON-DISTRIBUTION REGRESSION

## 3.1 MODEL SPECIFICATION

We consider the setting that we have access to a sample of $n$ independent and identically distributed covariate-response pairs of measures $\{(\mu_i, \nu_i)\}_{i=1}^{n}$ in $\mathcal{W}_2(\Omega) \times \mathcal{W}_2(\Omega)$. We consider the case where $\Omega$ is compact and assume without loss of generality that $\Omega = [0, 1]$. We let $P$ denote the joint distribution of the covariate and response measures $(\mu, \nu)$, and $P(\cdot|\mu)$ denote the conditional distribution of the response $\nu$ given the covariate $\mu$.

As in Ghodrati and Panaretos [2022], we define the regression operator $\Gamma : \mathcal{W}_2(\Omega) \to \mathcal{W}_2(\Omega)$ as the minimizer of the conditional Fréchet functional

$$\Gamma(\mu) := \text{argmin}_{b \in \mathcal{W}_2(\Omega)} \int_{\mathcal{W}_2(\Omega)} d_{\mathcal{W}}^2(b, \nu) dP(\nu|\mu), \quad (8)$$

where the definition above assumes the uniqueness of Fréchet mean of $P(\cdot|\mu)$. In a regression setting with covariate $x \in \mathbb{R}^d$ and scalar response $y \in \mathbb{R}$, the regression function $f : \mathbb{R}^d \to \mathbb{R}$ may be defined as $f(x) := \text{argmin}_{w \in \mathbb{R}} \mathbb{E}(|w - Y|^2 | X = x)$ where $\mathbb{E}(\cdot|X = x)$ conditional expectation of the response given covariate $x$. We note that in the formulation (8), the notion of expectation is replaced by a Wasserstein-Fréchet mean.

Ghodrati and Panaretos [2022] adopts a non-parametric approach and only impose a shape constraint on the regression operator by assuming $\Gamma(\mu) = T \# \mu$ where $T : \Omega \to \Omega$ is left unspecified and assumed to be a monotone increasing map.

In this work, we consider a parametric approach and model the map $T$ using the BP basis functions as described in Section 2.3:

$$T_{\boldsymbol{\theta}}(x) = \sum_{k=0}^{K} \theta_k G_{B(k,K-k+1)}(x),$$

where $\boldsymbol{\theta} = (\theta_0, \ldots, \theta_K)^T$ are the unknown coefficients. To ensure that $T_{\boldsymbol{\theta}}$ is monotonically increasing with range contained in $[0, 1]$, we require $\theta_k \geq 0$, $k = 0, \ldots, K$, and $\sum_{k=0}^{K} \theta_k = 1$.

We let $\Theta := \{\boldsymbol{\theta} \in \mathbb{R}^{K+1} : \theta_k \geq 0, k = 0, 1, \ldots, K, \sum_{k=0}^{K} \theta_k = 1\}$ denote the parameter space. We then assume that the regression operator $\Gamma$ satisfies $\Gamma(\mu) = T_{\boldsymbol{\theta}_0} \# \mu$ for some $\boldsymbol{\theta}_0 \in \Theta$. That is, we assume that the model is well-specified where the parameter space contains the "true" parameter $\boldsymbol{\theta}_0$. The optimal transport map $T_{\boldsymbol{\theta}_0}$ acts to move the probability mass assigned by the covariate measure $\mu$ from a subinterval

$(a, b) \subset \Omega$ to the corresponding transformed subinterval $(T_{\boldsymbol{\theta}_0}(a), T_{\boldsymbol{\theta}_0}(b))$.

While assuming that the parameter space $\Theta$ encompasses the true parameter $\boldsymbol{\theta}_0$ may seem somewhat limiting, our proof approach hinges on this assumption. This assumption is crucial to demonstrate the existence and uniqueness of the risk minimizer of (3) and enables us to establish the contraction rates of the generalized posterior distributions. Investigating the theoretical properties of the proposed framework when the proposed model does not contain the true optimal transport map is deferred to future research endeavors.

Given covariate-response pairs $(\mu_i, \nu_i)$, we assume the regression model takes the form

$$\nu_i = T_{\epsilon_i} \#(T_{\boldsymbol{\theta}_0} \# \mu_i), \quad i = 1, \ldots, n, \qquad (9)$$

where $T_{\epsilon_i}$ are independent and identically distributed random transport maps. $T_{\epsilon_i}$ can be interpreted as noise in the model. As in Ghodrati and Panaretos [2022], the specific distribution of $T_{\epsilon_i}$ is left unspecified, and is only required to be monotonically increasing and satisfy $\mathbb{E}(T_{\epsilon_i}(x)) = x$ for almost every $x \in \Omega$.

Hence, our model structure can be understood as a semi-parametric approach, comprising a parametric component for the optimal transport map $T_{\boldsymbol{\theta}}$ and a nonparametric component for the random error maps $T_{\epsilon_i}$. In contrast to a conventional Bayesian framework that necessitates fully specifying the random error maps, in the generalized Bayesian setting, they can remain unspecified. This offers several advantages. Firstly, fully parameterizing the random error maps might prove challenging and increase the likelihood of model mis-specification. Secondly, by leaving the random error maps unspecified, we can concentrate our modeling efforts on the optimal transport map $T_{\boldsymbol{\theta}}$, resulting in more efficient posterior sampling.

With the modeling assumptions described above, the distribution-on-distribution regression problem now becomes inferring the unknown parameter $\boldsymbol{\theta}_0 = (\theta_{0,0}, \ldots, \theta_{0,K})^T$ from a sample of covariate-response measures $\{\mu_i, \nu_i\}_{i=1}^n$. Choosing the 2-Wasserstein distance (1) as the loss function

$$\ell_{\boldsymbol{\theta}}(\mu, \nu) = \frac{1}{2} d_{\mathcal{W}}^2(T_{\boldsymbol{\theta}} \# \mu, \nu),$$

the expected risk is given by

$$R(\boldsymbol{\theta}) = \frac{1}{2} \int_{\mathcal{W}_2(\Omega) \times \mathcal{W}_2(\Omega)} d_{\mathcal{W}}^2(T_{\boldsymbol{\theta}} \# \mu, \nu) dP(\mu, \nu). \quad (10)$$

We first state two assumptions to ensure that the true parameter $\boldsymbol{\theta}_0$ is the unique parameter which minimizes the expected risk (10). These assumptions are analogous to

those in Ghodrati and Panaretos [2022].

Let $P_M$ be the marginal distribution of the covariate measure $\mu$.

**Assumption 1.** *Let $\mu$ be in the support of $P_M$, then $\mu$ is absolutely continuous with respect to the Lebesgue measure on $\Omega$.*

**Assumption 2.** *The true regression model has the form $\nu = T_\epsilon \#(T_{\boldsymbol{\theta}_0} \# \mu)$ for some $\boldsymbol{\theta}_0 \in \Theta$, and the random optimal transport map $T_\epsilon$ satisfies $\mathbb{E}(T_\epsilon(x)) = x$ $\Omega$-a.e.*

**Proposition 1.** *Suppose that the joint distribution $P$ induced by the model (9) satisfies Assumptions 1 and 2. Then $\boldsymbol{\theta}_0$ is the unique minimizer of the expected risk in (10).*

*Proof.* This result is a direct consequence of Theorem 3.3 of Ghodrati and Panaretos [2022]. □

## 3.2 PRIOR SPECIFICATION

We now specify the prior on the unknown coefficients $\boldsymbol{\theta}$. Our prior structure is similar to the one adopted by Curtis and Ghosh [2011]. We first sample $K + 1$ binary latent indicator random variables with parameter $p_\gamma$:

$$\gamma_0, \ldots, \gamma_K \sim \text{Ber}(p_\gamma).$$

In particular, $p_\gamma$ determines the sparsity of the binary variables $\gamma_k, k = 0, 1, \ldots, K$ We assign a beta prior on $p_\gamma$:

$$p_\gamma \sim \text{Be}(a_p, b_p), \quad a_p > 0, b_p > 0.$$

Conditional on $\gamma_0, \ldots, \gamma_K$, we sample $u_k$ as

$$u_k \sim \gamma_k \text{Unif}(0, 1) + (1 - \gamma_k)\delta_{\{0\}}, \quad k = 0, \ldots, K, \quad (11)$$

where $\text{Unif}(0, 1)$ is the uniform distribution on $(0, 1)$ and $\delta_{\{0\}}$ is the Dirac measure on 0. Finally, if $\sum_{k=0}^K \gamma_k > 0$, we set

$$\theta_k = \frac{u_k}{\sum_{j=0}^K u_j}, \quad k = 0, \ldots, K.$$

Otherwise, if $\sum_{k=0}^K \gamma_k = 0$, we set $\theta_k = 0$, $k = 0, \ldots, K$. In particular, the case $\theta_k = 0$ for all $k$ corresponds to the transport map $T(x) = 0$ for all $x \in [0, 1]$. This is a degenerate case where the covariate measure $\mu$ is transformed to the Dirac measure $\nu = \delta_{(0)}$.

**Alternative Prior Specification**
If prior information about the shape of the transport map is available, one may incorporate this information in the prior specification. Instead of sampling the random variables $u_k$ as a mixture of uniform distribution and Dirac measure $\delta_{(0)}$ as in (11), we instead sample $u_k$ as

$$u_k \sim \gamma_k \text{Beta}(a_k, b_k) + (1 - \gamma_k)\delta_{(0)},$$

for appropriately chosen values $a_k, b_k > 0, k = 0, \ldots, K$. We note that (11) is recovered by setting $a_k = 1, b_k = 1$.

## 3.3 CONCENTRATION OF POSTERIOR DISTRIBUTION

Building upon the model assumptions detailed in Section 3.1 and the prior specification presented in Section 3.2, we study the contraction rates of the Gibbs posterior distribution in two scenarios. The first scenario involves perfect observation of both covariate and response measures, while the second scenario entails solely observing samples from the respective covariate and response measures.

### 3.3.1 Perfect Observations

We first consider the case where the measures $\{(\mu_i, \nu_i)\}_{i=1}^n$ are perfectly observed. The empirical risk corresponding to the expected risk in (10) is given by

$$
\begin{aligned}
R_n(\boldsymbol{\theta}) &:= \frac{1}{n}\sum_{i=1}^n \ell_{\boldsymbol{\theta}}(\mu_i, \nu_i) \\
&= \frac{1}{2n}\sum_{i=1}^n d_{\mathcal{W}}^2(T_{\boldsymbol{\theta}}\#\mu_i, \nu_i). \quad (12)
\end{aligned}
$$

Before stating our first theoretical result, we have to introduce a distance on the space of optimal transport maps $\{T_{\boldsymbol{\theta}} : \boldsymbol{\theta} \in \Theta\}$. As in Ghodrati and Panaretos [2022], we measure the distance between two optimal transport maps using the $L^2(Q)$ distance where $Q$ is the measure defined as the linear average of $P_M$:

$$
Q(A) = \int_{\mathcal{W}_2(\Omega)} \mu(A) dP_M(\mu), \quad A \subset \Omega.
$$

We show that the Gibbs posterior resulting from the empirical risk in (12) and the prior distribution specified in Section 3.2 contracts around the true optimal map $T_{\boldsymbol{\theta}_0}$ with respect to $||\cdot||_{L^2(Q)}$ at rate (at least) $\epsilon_n = n^{-1/2}(\log n)^{1/2}$.

**Theorem 1.** *Suppose Assumption 1 and 2 hold. The Gibbs posterior distribution* (4) *with empirical risk in* (12) *asymptotically concentrates around the true optimal transport map $T_{\boldsymbol{\theta}_0}$ where $\boldsymbol{\theta}_0$ is the unique minimizer of $R(\boldsymbol{\theta})$ defined in* (10) *with respect to $||\cdot||_{L^2(Q)}$ at rate (at least) $\epsilon_n = n^{-1/2}(\log n)^{1/2}$. That is,*

$$
P^n\Pi_n(\{\boldsymbol{\theta} : ||T_{\boldsymbol{\theta}} - T_{\boldsymbol{\theta}_0}||_{L^2(Q)} > M\epsilon_n\}) \to 0
$$

*as $n \to \infty$.*

It's worth noting that the contraction result is expressed in terms of the optimal transport map $T_{\boldsymbol{\theta}}$, while the prior distribution is assigned to the parameter $\boldsymbol{\theta}$. The Gibbs posterior will concentrate around $T_{\boldsymbol{\theta}_0}$ as long as it concentrates around $\boldsymbol{\theta}_0$ since

$$
||T_{\boldsymbol{\theta}} - T_{\boldsymbol{\theta}_0}||_{L^2(Q)} \lesssim ||\boldsymbol{\theta} - \boldsymbol{\theta}_0||_2,
$$

where $||\cdot||_2$ is the $L^2$ norm on $\mathbb{R}^{K+1}$.

The proof of Theorem 1 is provided in the supplementary material. In particular, the proof applies Theorem 3.2 of Syring and Martin [2023]. This amounts to verifying a sub-exponential condition on the loss function and that the prior probability measure puts sufficient amount of mass around certain "neighborhood" of the true parameter $\boldsymbol{\theta}_0$.

As our loss function $\ell_{\boldsymbol{\theta}}(\mu, \nu)$ is bounded with respect to both the parameter $\boldsymbol{\theta}$ and the measures $(\mu, \nu)$, we can employ the approach outlined in Section 3.4.1 of Syring and Martin [2023] to verify the condition concerning the loss function. Since the parameter space is finite dimensional, the prior mass condition can be easily verified.

The presence of the logarithm factor in the contraction rate is due to the fact that in the finite dimensional case, it is impossible for a fixed prior to assign mass bounded away from 0 to a shrinking neighborhood of $\boldsymbol{\theta}_0$.

We note that the proof of the contraction result in Theorem 1 does not rely on the properties of Bernstein basis functions. Therefore, this result can be adapted to alternative choices of basis functions, such as monotone B-spline bases [Leitenstorfer and Tutz, 2006].

### 3.3.2 Imperfect Observations

We now consider the case where we do not observe $\{(\mu_i, \nu_i)\}_{i=1}^n$ but rather samples $\{x_{ij}\}_{j=1}^m$ and $\{y_{ij}\}_{j=1}^m$ from $\mu_i, \nu_i$, respectively, for $i = 1, \ldots, n$. We let $\hat{\mu}_i^m$ denote the estimated covariate measure $\mu_i$ based on the sample $\{x_{ij}\}_{j=1}^m$ and $\hat{\nu}_i^m$ denote the estimated response measure $\nu_i$ based on the sample $\{y_{ij}\}_{j=1}^m$. For simplicity we consider the setting where the sample size $m$ is the same for all covariate and response measures.

To study the rate of convergence of the Gibbs posterior distribution, we assume that for any $(\mu, \nu) \sim P$, we have a sequence of (deterministic) absolutely continuous measures $\hat{\mu}^m$ and a sequence of (deterministic) measures $\hat{\nu}^m$ such that

$$
\begin{aligned}
d_{\mathcal{W}}(\hat{\mu}^m, \mu) &\lesssim r_m^{-1} \\
d_{\mathcal{W}}(\hat{\nu}^m, \nu) &\lesssim r_m^{-1},
\end{aligned}
$$

where $r_m^{-1}$ is the convergence rate with respect to the 2-Wasserstein distance.

The empirical risk in this setting is given by

$$
\begin{aligned}
\tilde{R}_n^m(\boldsymbol{\theta}) &:= \frac{1}{n}\ell_{\boldsymbol{\theta}}(\hat{\mu}^m, \hat{\nu}^m) \\
&= \frac{1}{2n}\sum_{i=1}^n d_{\mathcal{W}}^2(T_{\boldsymbol{\theta}}\#\hat{\mu}_i^m, \hat{\nu}_i^m). \quad (13)
\end{aligned}
$$

In order for the Gibbs posterior distribution to contract around the true optimal map $T_{\boldsymbol{\theta}_0}$, we assume that $m(n)$ is a deterministic function of $n$ and that $m \to \infty$ as $n \to \infty$ and $r_m^{-1} \to 0$ as $m \to \infty$, and require that $r_m^{-\frac{1}{2}} < \frac{1}{2} n^{-\frac{1}{2}} \log n$ for all $n$.

**Theorem 2.** *Suppose Assumption 1 and 2 hold. Suppose $r_m^{-\frac{1}{2}} < \frac{1}{2} n^{-\frac{1}{2}} \log n$ for all $n$. The Gibbs posterior distribution* (4) *with empirical risk in* (13) *asymptotically concentrates around the true optimal transport map $T_{\boldsymbol{\theta}_0}$ where $\boldsymbol{\theta}_0$ is the unique minimizer of $R(\boldsymbol{\theta})$ defined in* (10) *with respect to $\| \cdot \|_{L^2(Q)}$ at rate (at least) $\epsilon_n \asymp n^{-1/2}(\log n)^{1/2}$. That is,*

$$P^n \Pi_n(\{\boldsymbol{\theta} : \|T_{\boldsymbol{\theta}} - T_{\boldsymbol{\theta}_0}\|_{L^2(Q)} > M\epsilon_n\}) \to 0$$

*as $n \to \infty$.*

The proof of Theorem 2 is presented in the supplementary material. Directly applying the general contraction theorems presented in Syring and Martin [2023] is not feasible in the current context. Nonetheless, we can modify the proof methodology in Theorem 3.2 of Syring and Martin [2023] to suit our current scenario.

## 3.4 BAYESIAN COMPUTATION

With prior specification described above, we sample from the Gibbs posterior distribution of $u_0, \ldots, u_K, \gamma_0, \ldots, \gamma_K$ and $p_\gamma$. The posterior of $\boldsymbol{\theta}$ and hence the optimal transport map $T_{\boldsymbol{\theta}}$ are then induced from the posterior of $u_0, \ldots, u_K$. We describe the sampling procedure for the scenario of perfect observation of measures, highlighting that the process for imperfect observation is entirely analogous.

We first initialize all parameters $(\gamma_0^{(0)}, \gamma_1^{(0)}, \ldots, \gamma_K^{(0)})$, $p_\gamma^{(0)}$, and $(u_0^{(0)}, u_1^{(0)}, \ldots, u_K^{(0)})$. For each iteration $t+1$, and for each $k = 0, 1, \ldots, K$, we jointly sample $(\gamma_k^{(t+1)}, u_k^{(t+1)})$ as follows:

$$\tilde{\gamma}_k^{(t+1)} = \begin{cases} 1 - \gamma_k^{(t)} & \text{with probability} = q_\gamma \\ \gamma_k^{(t)} & \text{with probability} = 1 - q_\gamma, \end{cases}$$

where $q_\gamma \in (0,1)$.

Conditional on $\tilde{\gamma}_k^{(t+1)} = 0$, we set $\tilde{u}_k^{(t+1)} = 0$. Otherwise, we draw $\tilde{u}_k^{(t+1)}$ from $\text{Unif}(0,1)$.

We then compute $\tilde{\boldsymbol{\theta}}^{(t+1)}$ using

$$u_0^{(t+1)}, u_1^{(t+1)}, \ldots, u_{k-1}^{(t+1)}, \tilde{u}_k^{(t+1)}, u_{k+1}^{(t)}, \ldots, u_K^{(t)},$$

and set $(\gamma_k^{(t+1)}, u_k^{(t+1)}) = (\tilde{\gamma}_k^{(t+1)}, \tilde{u}_k^{(t+1)})$ with probability equal to

$$\min \left\{ 1, \frac{e^{-\omega n R_n(\tilde{\boldsymbol{\theta}}^{(t+1)})} \pi_\gamma(\tilde{\gamma}_k^{(t+1)})}{e^{-\omega n R_n(\boldsymbol{\theta}^{(t)})} \pi_\gamma(\gamma_k^{(t)})} \right\}, \qquad (14)$$

and $(\gamma_k^{(t+1)}, u_k^{(t+1)}) = (\gamma_k^{(t)}, u_k^{(t)})$ otherwise, and $\pi_\gamma(\gamma_k) = p_\gamma^{\gamma_k}(1 - p_\gamma)^{1-\gamma_k}$.

The full conditional posterior distribution of $p_\gamma$ is of closed form and can be sampled directly:

$$p_\gamma^{(t+1)} \sim \text{Ber}\left(a_p + \sum_{k=0}^{K} \gamma_k^{(t+1)}, b_p + K + 1 - \sum_{k=0}^{K} \gamma_k^{(t+1)}\right).$$

We need to specify the polynomial order $K$ for BP. Unlike monotone regression, where various strategies exist for determining $K$, such as setting it to the order of unique predictor values in the data, our context is more intricate. Therefore, we opt for a larger value of $K$ and rely on the fitting procedure to eliminate unnecessary basis functions. In our simulation studies and data application, we opt for $K = 50$ while also examining alternative values for $K$.

## 4 SIMULATION STUDIES

We conduct simulation studies to investigate the concentration of the generalized posterior distributions around the true optimal transport maps. In order to carry out these simulation studies, we need to simulate the covariate probability measures $\{\mu_i\}_{i=1}^n$, the optimal transport map $T_{\boldsymbol{\theta}_0}$, and the random error maps $\{T_{\epsilon_i}\}_{i=1}^n$.

Each of the covariate measures is assumed to be Beta distribution $\text{Be}(a, b)$, with the parameters $a$ and $b$ generated randomly from beta distributions, where $a \sim \text{Be}(a_1, b_1)$ and $b \sim \text{Be}(a_2, b_2)$.

The parameters $\boldsymbol{\theta}_0$ governing the true optimal transport map $T_{\boldsymbol{\theta}_0}$ are sampled from the prior distribution outlined in Section 3.2. In all simulations, we fix $K = 50$, while the sparsity parameter $p_\gamma$ varies across different simulation scenarios.

To generate the random error maps $\{T_{\epsilon_i}\}_{i=1}^n$, we consider the class of random error optimal maps introduced in Panaretos and Zemel [2016]. The maps are defined as

$$\Psi_0(x) = x,$$

$$\Psi_z(x) = x - \frac{\sin(\pi z x)}{|z|\pi}, \quad z \in \mathbb{Z} - \{0\}.$$

These are strictly increasing smooth functions satisfying $\Psi_z(0) = 0$ and $\Psi_z(1) = 1$ for all $z \in \mathbb{Z}$. Random maps can be constructed by replacing the integer $z$ with a random variable $Z$ that has a distribution symmetric around 0, it is straightforward to see that $\mathbb{E}(\Psi_Z(x)) = x$ for all $x \in [0,1]$.

As in Panaretos and Zemel [2016], we use the following distribution for $Z$ parameterized by $\lambda > 0$:

$$\mathbb{P}(Z = 0) = e^{-\lambda},$$

$$\mathbb{P}(Z = +z) = \mathbb{P}(Z = -z) = \frac{e^{-\lambda}\lambda^z}{2(z!)}, \quad z \in \mathbb{Z} - \{0\}.$$

The random error map can be constructed as follows: for $J > 1$, we simulate i.i.d. integer-valued symmetric random variables $z_j, j = 1, \ldots, J$. We then simulate $J - 1$ uniform random variables $v_1, \ldots, v_{J-1} \sim \text{Unif}(0, 1)$. We let $v_{(1)}, \ldots, v_{(J-1)}$ denote the order statistics of $v_1, \ldots, v_{J-1}$. The random error map is then given by

$$
\begin{aligned}
T_\epsilon(x) &= v_{(1)}\Psi(x) + \sum_{j=2}^{J-1}(v_{(j)} - v_{(j-1)})\Psi_{z_j}(x) \\
&\quad + (1 - v_{(J-1)})\Psi_{z_J}(x).
\end{aligned}
$$

In all of our simulation scenarios, we set $J = 20$ and $\lambda = 5$. The response measures $\{\nu_i\}_{i=1}^n$ are then generated based on the generated covariate measures $\{\mu_i\}_{i=1}^n$, the optimal transport map $T_{\boldsymbol{\theta}_0}$, and the random error maps $\{T_{\epsilon_i}\}_{i=1}^n$.

We carry out three simulation studies, producing $n = 100$ pairs of covariate-response measures for each study using the method outlined previously. Then, we perform posterior sampling on the simulated data, following the procedure outlined in Section 3.4. In each simulation study, we apply the algorithm described in Section 3.4 to subsets of the data containing $n = 5$, $n = 20$, $n = 50$, and $n = 100$ pairs of covariate-response measures to explore how the behavior of the posterior distribution varies with sample size.

The determination of the learning rate $\omega$ for the generalized posterior distribution is crucial. Several strategies for selecting $\omega$ have been proposed [Grünwald and van Ommen, 2017, Holmes and Walker, 2017, Lyddon et al., 2019, Syring and Martin, 2018], with a comparative analysis presented in Wu and Martin [2023]. In our current scenario, we adopt the strategies of Syring and Martin [2018] and Syring and Martin [2023] to let $\omega$ be a decreasing function of the sample size $n$ and also tune $\omega$ to ensure the resulting Gibbs posterior has sufficient coverage probability.

The outcomes of the simulation studies are depicted in Figures 1, 2, and 3. These figures showcase the estimated posterior means and posterior credible intervals of the optimal transport map, juxtaposed with the true optimal transport map. Notably, the posterior mean of the optimal transport map adeptly captures the true counterpart across all scenarios, with the true map being well-contained within the posterior credible intervals. These observations hold true across all three simulation scenarios and various sample sizes. Hence, even with a smaller sample size of $n = 5$, the recovery of the true optimal transport map appears successful. This observation is not entirely unexpected, as one can interpret each covariate-response pair as containing an infinite amount of data points.

In the supplementary material, we will further investigate the behavior of the model when it is misspecified for the true optimal transport maps. The first scenario involves the specified models having fewer basis functions $K$ than the true optimal transport maps. The second, more challenging scenario occurs when the true optimal transport maps do not belong to the model class, meaning they cannot be expressed as linear combinations of BP basis functions.

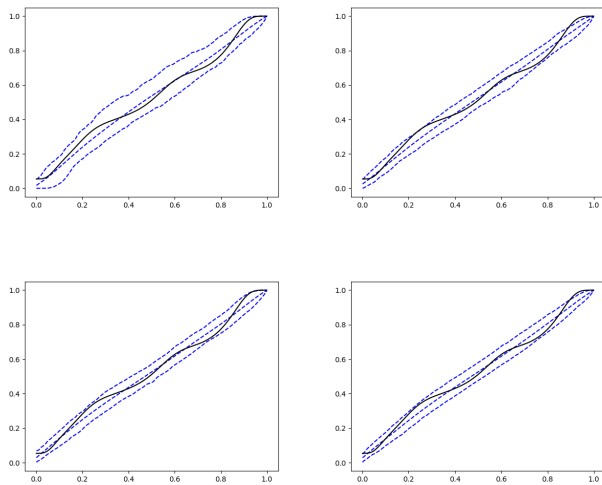

Figure 1: Simulation 1. Top left: n=5 ($\omega = 500$). Top right: n=20 ($\omega = 200$). Bottom left: n=50 ($\omega = 100$). Bottom right: n=100 ($\omega = 50$). Black curve: True optimal transport map. Blue dashed curves: estimated posterior mean and 95% posterior credible intervals of optimal transport map.

# 5 DATA APPLICATION - ANALYSIS OF MORTALITY DATA

We examine the age-at-death distributions for $N = 37$ countries in the years 1983 and 2013, sourced from the Human Mortality Database accessible via UC Berkeley and the Max Planck Institute for Demographic Research, openly available on www.mortality.org. The provided death rates span single years of age up to 109, with an open age interval for individuals aged 110 and above. Utilizing the binsmooth R package (version 0.2.2), we fit smooth cubic splines to the binned data to estimate the cumulative distribution functions (CDFs) for each country for both year 1983 and 2013. Specifically, we designate the age-at-death distribution for the $i$th country in the year 1983 as the covariate distribution, and the corresponding distribution for the same country in the year 2013 as the response distribution. This allows for comparisons with the studies by Pegoraro and Beraha [2022] and Chen et al. [2023].

Figure 4 displays the posterior mean of the optimal

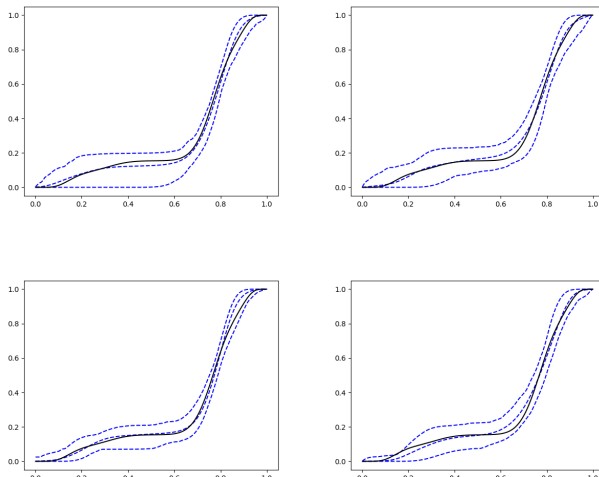

Figure 2: Simulation 2. Top left: n=5 ($\omega = 500$). Top right: n=20 ($\omega = 200$). Bottom left: n=50 ($\omega = 100$). Bottom right: n=100 ($\omega = 50$). Black curve: True optimal transport map. Blue dashed curves: estimated posterior mean and 95% posterior credible intervals of optimal transport map.

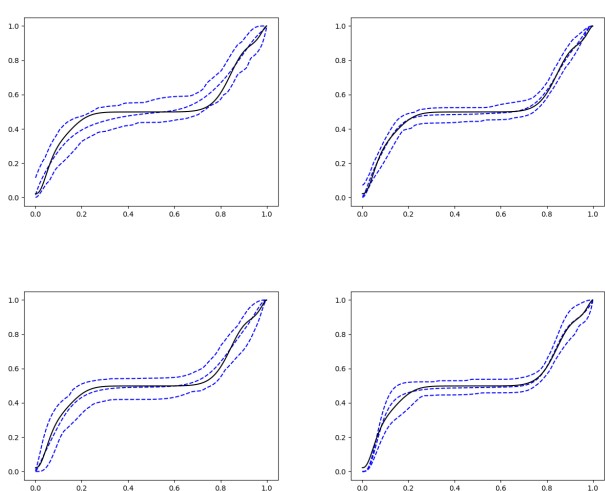

Figure 3: Simulation 3. Top left: n=5 ($\omega = 500$). Top right: n=20 ($\omega = 200$). Bottom left: n=50 ($\omega = 100$). Bottom right: n=100 ($\omega = 50$). Black curve: True optimal transport map. Blue dashed curves: estimated posterior mean and 95% posterior credible intervals of optimal transport map.

transport map, accompanied by 95% credible intervals. These results are derived through the MCMC sampling approach outlined in Section 3.4, utilizing covariate-response pairs for the $n = 37$ countries. Here, we set $K = 50$ and $p_\gamma = 0.2$. Notably, the estimated posterior mean optimal transport map surpasses the identity transport map pointwise, suggesting an overall enhancement in mortality

rates across all age groups, with the most significant improvements observed in younger age groups. These findings align with those of Pegoraro and Beraha [2022]. To evaluate the model's goodness of fit, we compare the cumulative distribution functions (CDFs) at year 2013 obtained from the MCMC samples with the observed CDFs at year 2013. Figure 5 illustrates this comparison, indicating a favorable fit of the model to the data.

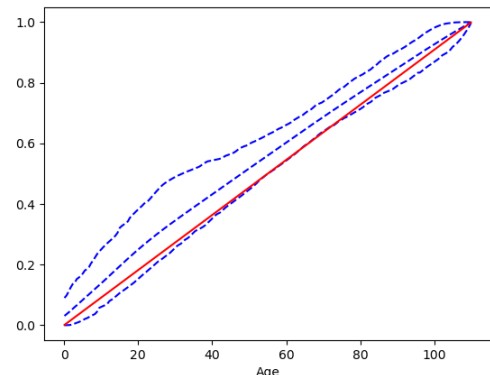

Figure 4: Mortality dataset. Blue: posterior mean and 95% credible intervals of the optimal transport map. Red: identity transport map.

# 6 DISCUSSION

In this study, we introduced a generalized Bayesian framework for distribution-on-distribution regression. We studied the contraction rates of the Gibbs posterior distribution under two scenarios: one where both covariate and response measures are fully observed, and another where we only have access to samples from these measures. Experimental studies were conducted to investigate the contraction properties of the posterior distribution.

In our theoretical analysis, we made the assumption that the true optimal transport map can be represented as convex combinations of basis functions from BP. However, it is desirable to examine the contraction rates of the posterior distribution in cases where the true optimal transport map does not conform to this assumed form. This may necessitate a different proof strategy, as our current approach heavily relies on this assumption. Additionally, an intriguing extension would involve extending the framework to higher dimensional settings. Nonetheless, in higher dimensions, there's often a trade-off between the flexibility of optimal transport maps and computational efficiency. One potential avenue to explore is adapting the additive monotone regression approach proposed by Engebretsen and Glad [2019] to our present context.

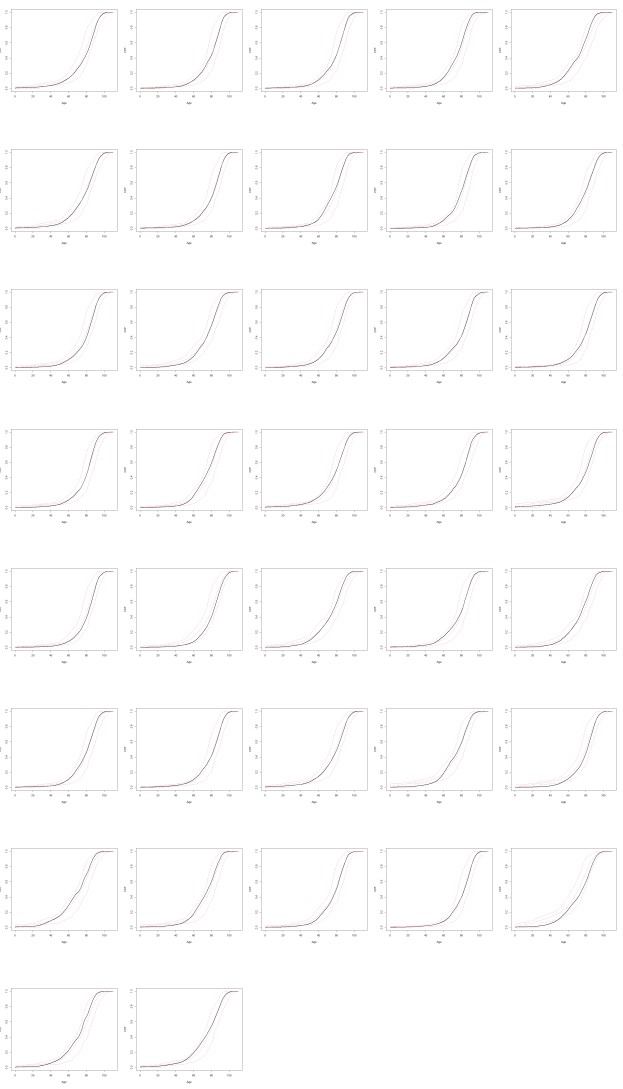

Figure 5: Black: True CDFs of age-at-death distribution in year 2013 for each country. Red: Estimated posterior means and posterior credible intervals of CDFs in year 2013 for each country.

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

# A Generalized Bayesian Approach to Distribution-on-Distribution Regression (Supplementary Material)

**Tin Lok James Ng**[1]

[1]School of Computer Science and Statistics, Trinity College Dublin, Dublin, Ireland

# A PROOFS

## A.1 PROOF OF THEOREM 1

We first state a lemma which is a direct implication of Lemma 3.6 of Ghodrati and Panaretos [2022]. For $\boldsymbol{\eta} \in \mathbb{R}^{K+1}$, we define

$$T_{\boldsymbol{\eta}}(x) := \sum_{k=0}^{M} \eta_k G_{B(k, K-1+k)}(x).$$

Note that we have extended the definition of the map $T_{\boldsymbol{\eta}}$ from $\Theta$ to $\mathbb{R}^{K+1}$.

**Lemma 1.** *For $\epsilon > 0$, we have the following expansion of the expected risk around $\boldsymbol{\theta}_1$:*

$$R(\boldsymbol{\theta}_1 + \epsilon\boldsymbol{\eta}) = R(\boldsymbol{\theta}_1) + \epsilon D_{\boldsymbol{\eta}} R(\boldsymbol{\theta}_1) + \frac{\epsilon^2}{2} ||T_{\boldsymbol{\eta}}||_{L^2(Q)}^2$$

*where*

$$D_{\boldsymbol{\eta}} R(\boldsymbol{\theta}_1) = \int_{\mathcal{W}_2(\Omega) \times \mathcal{W}_2(\Omega)} \int_0^1 (T_{\boldsymbol{\theta}}(F_\mu^{-1}(p)) - F_\nu^{-1}(p)) T_{\boldsymbol{\eta}}(F_\mu^{-1}(p)) dp \, dP(\mu, \nu)$$

*is the directional derivative of $R(\boldsymbol{\theta}_1)$ in the direction of $\boldsymbol{\eta}$.*

*Proof of Theorem 1.* We apply Theorem 3.2 of Syring and Martin [2023] to derive the stated contraction rate.

We need to show that the loss function $\ell_{\boldsymbol{\theta}}$ satisfies the sub-exponential condition:

There exists an interval $(0, \bar{\omega})$ and constant $K > 0$ such that for all $\omega \in (0, \bar{\omega})$ and for all sufficiently small $\delta > 0$, for $\boldsymbol{\theta} \in \Theta$,

$$||T_{\boldsymbol{\theta}} - T_{\boldsymbol{\theta}_0}||_{L^2(Q)} > \delta \implies Pe^{-\omega(\ell_{\boldsymbol{\theta}} - l_{\boldsymbol{\theta}_0})} < e^{-K\omega\delta^2}. \tag{15}$$

We also need to show that the prior $\Pi$ puts sufficient amount of mass on "neighborhood" $G_n$ of the true parameter $\boldsymbol{\theta}_0$:

$$\log \Pi(G_n) \gtrsim -n\epsilon_n^2, \tag{16}$$

where $G_n$ is defined as

$$G_n := \{\boldsymbol{\theta} \in \Theta : u(\boldsymbol{\theta}, \boldsymbol{\theta}_0) \le \epsilon_n^2, v(\boldsymbol{\theta}, \boldsymbol{\theta}_0) \le \epsilon_n^2\}, \quad n = 1, 2, \dots,$$

and $u(\boldsymbol{\theta}, \boldsymbol{\theta}_0)$ and $v(\boldsymbol{\theta}, \boldsymbol{\theta}_0)$ are the mean and variance of excess risk:

$$u(\boldsymbol{\theta}, \boldsymbol{\theta}_0) := \frac{1}{2} P(d_{\mathcal{W}}^2(T_{\boldsymbol{\theta}} \# \mu, \nu) - d_{\mathcal{W}}^2(T_{\boldsymbol{\theta}_0} \# \mu, \nu)) = R(\boldsymbol{\theta}) - R(\boldsymbol{\theta}_0),$$

and

$$v(\boldsymbol{\theta}, \boldsymbol{\theta}_0) := P\left(\left(\frac{1}{2}d_{\mathcal{W}}^2(T_{\boldsymbol{\theta}}\#\mu, \nu) - \frac{1}{2}d_{\mathcal{W}}^2(T_{\boldsymbol{\theta}_0}\#\mu, \nu)\right)^2\right) - u(\boldsymbol{\theta}, \boldsymbol{\theta}_0)^2.$$

We first show that the sub-exponential condition (15) is satisfied. By compactness of $\Omega$, we have that for all $\boldsymbol{\theta} \in \Theta$ and all $(\mu, \nu)$ in the support of $P$,

$$\ell_{\boldsymbol{\theta}}(\mu, \nu) - \ell_{\boldsymbol{\theta}_0}(\mu, \nu) < C,$$

for some constant $C > 0$. Thus, by Section 3.4.1 of Syring and Martin [2023],

$$Pe^{-\omega(\ell_{\boldsymbol{\theta}} - l_{\boldsymbol{\theta}_0})} \leq \exp\left(-\omega u(\boldsymbol{\theta}, \boldsymbol{\theta}_0) + C\omega^3 v(\boldsymbol{\theta}, \boldsymbol{\theta}_0)\right),$$

for $\omega$ small enough.

Now, consider $\boldsymbol{\theta}_1, \boldsymbol{\theta}_2 \in \Theta$, and let $\boldsymbol{\eta} = \boldsymbol{\theta}_2 - \boldsymbol{\theta}_1$. By Lemma 1, we have

$$R(\boldsymbol{\theta}_1 + \epsilon\boldsymbol{\eta}) = R(\boldsymbol{\theta}_1) + \epsilon D_{\boldsymbol{\eta}}R(\boldsymbol{\theta}_1) + \frac{\epsilon^2}{2}\|T_{\boldsymbol{\eta}}\|_{L^2(Q)}^2$$

where $D_{\boldsymbol{\eta}}R(\boldsymbol{\theta}_1)$ is the directional derivative of $R(\boldsymbol{\theta}_1)$ in the direction of $\boldsymbol{\eta}$.

For any $\boldsymbol{\theta} \in \Theta$, let $\boldsymbol{\eta} = \boldsymbol{\theta} - \boldsymbol{\theta}_0$, applying the expansion above with $\epsilon = 1$, we have

$$R(\boldsymbol{\theta}) - R(\boldsymbol{\theta}_0) = D_{\boldsymbol{\eta}}R(\boldsymbol{\theta}_0) + \frac{1}{2}\|T_{\boldsymbol{\eta}}\|_{L^2(Q)}^2.$$

Since $\boldsymbol{\theta}_0$ is the minimizer of $R$, we have $D_{\boldsymbol{\eta}}R(\boldsymbol{\theta}_0) = 0$, and thus

$$u(\boldsymbol{\theta}, \boldsymbol{\theta}_0) = R(\boldsymbol{\theta}) - R(\boldsymbol{\theta}_0) = \frac{1}{2}\|T_{\boldsymbol{\eta}}\|_{L^2(Q)}^2 = \frac{1}{2}\|T_{\boldsymbol{\theta}} - T_{\boldsymbol{\theta}_0}\|_{L^2(Q)}^2. \tag{17}$$

We also have that

$$\begin{aligned}
|d_{\mathcal{W}}^2(T_{\boldsymbol{\theta}_1}\#\mu, \nu) - d_{\mathcal{W}}^2(T_{\boldsymbol{\theta}_2}\#\mu, \nu)| &\lesssim |d_{\mathcal{W}}(T_{\boldsymbol{\theta}_1}\#\mu, \nu) - d_{\mathcal{W}}(T_{\boldsymbol{\theta}_2}\#\mu, \nu)| \\
&\leq d_{\mathcal{W}}(T_{\boldsymbol{\theta}_1}\#\mu, T_{\boldsymbol{\theta}_2}\#\mu) \\
&= \|T_{\boldsymbol{\theta}_1} - T_{\boldsymbol{\theta}_2}\|_{L^2(\mu)} \\
&\lesssim \|T_{\boldsymbol{\theta}_1} - T_{\boldsymbol{\theta}_2}\|_{L^2(Q)}.
\end{aligned}$$

where the second inequality follows from triangle inequality, and the equality follows from that $\mathcal{W}_2(\Omega)$ is flat. Therefore, it follows that

$$v(\boldsymbol{\theta}, \boldsymbol{\theta}_0) \lesssim \|T_{\boldsymbol{\theta}} - T_{\boldsymbol{\theta}_0}\|_{L^2(Q)}^2 \lesssim u(\boldsymbol{\theta}, \boldsymbol{\theta}_0). \tag{18}$$

Combining (17) and (18), We obtain

$$\begin{aligned}
Pe^{-\omega(\ell_{\boldsymbol{\theta}} - l_{\boldsymbol{\theta}_0})} &\leq \exp\left(-C_1\omega u(\boldsymbol{\theta}, \boldsymbol{\theta}_0)\right) \\
&= \exp\left(-\frac{1}{2}C_1\omega\|T_{\boldsymbol{\theta}} - T_{\boldsymbol{\theta}_0}\|_{L^2(Q)}^2\right)
\end{aligned}$$

for some constant $C_1 > 0$. It follows that $\|T_{\boldsymbol{\theta}} - T_{\boldsymbol{\theta}_0}\|_{L^2(Q)} > \delta$ implies that

$$Pe^{-\omega(\ell_{\boldsymbol{\theta}} - l_{\boldsymbol{\theta}_0})} \leq \exp\left(-\frac{1}{2}C_1\omega\delta^2\right),$$

and Condition (15) is verified.

We now verify the prior mass condition (16). We note that our prior specification satisfies

$$\Pi(\{\|\boldsymbol{\theta} - \boldsymbol{\theta}_0\|_2 \leq \delta) \gtrsim \delta^{K+1},$$

where $||\cdot||_2$ is the 2-norm on $\mathbb{R}^{K+1}$. Since $||\boldsymbol{\theta} - \boldsymbol{\theta}_0||_2 \leq \delta$ implies $||T_{\boldsymbol{\theta}} - T_{\boldsymbol{\theta}_0}||_{L^2(Q)} \lesssim \delta$, it follows that

$$\Pi(\{||T_{\boldsymbol{\theta}} - T_{\boldsymbol{\theta}_0}||_{L^2(Q)} \leq \delta) \gtrsim \delta^{K+1}.$$

Since $||T_{\boldsymbol{\theta}} - T_{\boldsymbol{\theta}_0}||_{L^2(Q)} \leq \delta$ implies $\{u(\boldsymbol{\theta}, \boldsymbol{\theta}_0) \lesssim \delta^2, v(\boldsymbol{\theta}, \boldsymbol{\theta}_0) \lesssim \delta^2\}$, we have

$$\Pi(G_n) \gtrsim \Pi(\{\boldsymbol{\theta} : ||T_{\boldsymbol{\theta}} - T_{\boldsymbol{\theta}_0}||_{L^2(Q)} \leq \epsilon_n\}) \gtrsim \epsilon_n^{K+1}.$$

Therefore, with $\epsilon_n = n^{-1/2}(\log n)^{1/2}$, we have

$$\log \Pi(G_n) \gtrsim -\log n \gtrsim -n\epsilon_n^2.$$

Thus, the prior mass condition is satisfied, and the proof is completed. $\qquad\square$

## A.2 PROOF OF THEOREM 2

For each $m = 1, 2, \ldots$, we define

$$\tilde{R}^m(\boldsymbol{\theta}) := \frac{1}{2} \int_{\mathcal{W}_2(\Omega) \times \mathcal{W}_2(\Omega)} d_{\mathcal{W}}^2(T_{\boldsymbol{\theta}} \# \hat{\mu}^m, \hat{\nu}^m) dP(\mu, \nu). \tag{19}$$

Also define the mean and variance of excess risk as

$$u_m(\boldsymbol{\theta}, \boldsymbol{\theta}_0) := \frac{1}{2} P(d_{\mathcal{W}}^2(T_{\boldsymbol{\theta}} \# \hat{\mu}^m, \hat{\nu}^m) - d_{\mathcal{W}}^2(T_{\boldsymbol{\theta}_0} \# \hat{\mu}^m, \hat{\nu}^m)) = \tilde{R}^m(\boldsymbol{\theta}) - \tilde{R}^m(\boldsymbol{\theta}_0),$$

and

$$v_m(\boldsymbol{\theta}, \boldsymbol{\theta}_0) := P\left(\left(\frac{1}{2}d_{\mathcal{W}}^2(T_{\boldsymbol{\theta}} \# \hat{\mu}^m, \hat{\nu}^m) - \frac{1}{2}d_{\mathcal{W}}^2(T_{\boldsymbol{\theta}_0} \# \hat{\mu}^m, \hat{\nu}^m)\right)^2\right) - u_m(\boldsymbol{\theta}, \boldsymbol{\theta}_0)^2.$$

We first prove the following lemma bounding $u_m(\boldsymbol{\theta}, \boldsymbol{\theta}_0)$ and $v_m(\boldsymbol{\theta}, \boldsymbol{\theta}_0)$ in terms of $||T_{\boldsymbol{\theta}} - T_{\boldsymbol{\theta}_0}||_{L^2(Q)}^2$ and $r_m^{-1}$.

**Lemma 2.**

$$||T_{\boldsymbol{\theta}} - T_{\boldsymbol{\theta}_0}||_{L^2(Q)}^2 - r_m^{-1} \lesssim u_m(\boldsymbol{\theta}, \boldsymbol{\theta}_0) \lesssim ||T_{\boldsymbol{\theta}} - T_{\boldsymbol{\theta}_0}||_{L^2(Q)}^2 + r_m^{-1},$$

$$v_m(\boldsymbol{\theta}, \boldsymbol{\theta}_0) \lesssim ||T_{\boldsymbol{\theta}} - T_{\boldsymbol{\theta}_0}||_{L^2(Q)}^2 + r_m^{-2}.$$

*Proof.* We first have the following decomposition of $u_m(\boldsymbol{\theta}, \boldsymbol{\theta}_0)$:

$$\tilde{R}^m(\boldsymbol{\theta}) - \tilde{R}^m(\boldsymbol{\theta}_0) = \underbrace{\tilde{R}^m(\boldsymbol{\theta}) - R(\boldsymbol{\theta})} + \underbrace{R(\boldsymbol{\theta}) - R(\boldsymbol{\theta}_0)} + \underbrace{R(\boldsymbol{\theta}_0) - \tilde{R}^m(\boldsymbol{\theta}_0)}. \tag{20}$$

We bound each of the three terms on the RHS of (20).

$$\begin{aligned}
\tilde{R}^m(\boldsymbol{\theta}) - R(\boldsymbol{\theta}) &= P\left(d_{\mathcal{W}}^2(T_{\boldsymbol{\theta}} \# \hat{\mu}^m, \hat{\nu}^m) - d_{\mathcal{W}}^2(T_{\boldsymbol{\theta}} \# \mu, \nu)\right) \\
&= P\left(d_{\mathcal{W}}^2(T_{\boldsymbol{\theta}} \# \hat{\mu}^m, \hat{\nu}^m) - d_{\mathcal{W}}^2(T_{\boldsymbol{\theta}} \# \hat{\mu}^m, \nu) + d_{\mathcal{W}}^2(T_{\boldsymbol{\theta}} \# \hat{\mu}^m, \nu) - d_{\mathcal{W}}^2(T_{\boldsymbol{\theta}} \# \mu, \nu)\right).
\end{aligned}$$

We have that

$$\begin{aligned}
&P\left(d_{\mathcal{W}}^2(T_{\boldsymbol{\theta}} \# \hat{\mu}^m, \hat{\nu}^m) - d_{\mathcal{W}}^2(T_{\boldsymbol{\theta}} \# \hat{\mu}^m, \nu)\right) \\
&= P\left((d_{\mathcal{W}}(T_{\boldsymbol{\theta}} \# \hat{\mu}^m, \hat{\nu}^m) - d_{\mathcal{W}}(T_{\boldsymbol{\theta}} \# \hat{\mu}^m, \nu))(d_{\mathcal{W}}(T_{\boldsymbol{\theta}} \# \hat{\mu}^m, \hat{\nu}^m) + d_{\mathcal{W}}(T_{\boldsymbol{\theta}} \# \hat{\mu}^m, \nu))\right) \\
&\geq P\left(-d_{\mathcal{W}}(\hat{\nu}^m, \nu)(d_{\mathcal{W}}(T_{\boldsymbol{\theta}} \# \hat{\mu}^m, \hat{\nu}^m) + d_{\mathcal{W}}(T_{\boldsymbol{\theta}} \# \hat{\mu}^m, \nu))\right) \\
&\gtrsim P(-d_{\mathcal{W}}(\hat{\nu}^m, \nu)) \\
&\gtrsim -r_m^{-1},
\end{aligned}$$

where the first inequality follows from the reverse triangle inequality, and the last inequality follows from our assumption. We also have that

$$
\begin{aligned}
&P\left(d_\mathcal{W}^2(T_{\boldsymbol{\theta}}\#\hat{\mu}^m, \hat{\nu}^m) - d_\mathcal{W}^2(T_{\boldsymbol{\theta}}\#\hat{\mu}^m, \nu)\right) \\
=\ &P\left(\left(d_\mathcal{W}(T_{\boldsymbol{\theta}}\#\hat{\mu}^m, \hat{\nu}^m) - d_\mathcal{W}(T_{\boldsymbol{\theta}}\#\hat{\mu}^m, \nu)\right)\left(d_\mathcal{W}(T_{\boldsymbol{\theta}}\#\hat{\mu}^m, \hat{\nu}^m) + d_\mathcal{W}(T_{\boldsymbol{\theta}}\#\hat{\mu}^m, \nu)\right)\right) \\
\leq\ &P\left(d_\mathcal{W}(\hat{\nu}^m, \nu)\left(d_\mathcal{W}(T_{\boldsymbol{\theta}}\#\hat{\mu}^m, \hat{\nu}^m) + d_\mathcal{W}(T_{\boldsymbol{\theta}}\#\hat{\mu}^m, \nu)\right)\right) \\
\lesssim\ &r_m^{-1}
\end{aligned}
$$

by an application of the triangle inequality.

Similarly, we can show that

$$
-r_m^{-1} \lesssim P\left(d_\mathcal{W}^2(T_{\boldsymbol{\theta}}\#\hat{\mu}^m, \nu) - d_\mathcal{W}^2(T_{\boldsymbol{\theta}}\#\mu, \nu)\right) \lesssim r_m^{-1}.
$$

It follows that the first term on the RHS of (20) can be bounded as

$$
-r_m^{-1} \lesssim \tilde{R}^m(\boldsymbol{\theta}) - R(\boldsymbol{\theta}) \lesssim r_m^{-1}.
$$

Using the same calculation, we also have the bound for the third term on the RHS of (20):

$$
-r_m^{-1} \lesssim R(\boldsymbol{\theta}_0) - \tilde{R}^m(\boldsymbol{\theta}_0) \lesssim r_m^{-1}.
$$

For the second term on the RHS of (20), we recall from the proof of Theorem 1 that

$$
R(\boldsymbol{\theta}) - R(\boldsymbol{\theta}_0) = \frac{1}{2}\|T_{\boldsymbol{\theta}} - T_{\boldsymbol{\theta}_0}\|_{L^2(Q)}^2.
$$

Thus, we obtain the following bound for $u_m(\boldsymbol{\theta}, \boldsymbol{\theta}_0)$:

$$
\|T_{\boldsymbol{\theta}} - T_{\boldsymbol{\theta}_0}\|_{L^2(Q)}^2 - r_m^{-1} \lesssim u_m(\boldsymbol{\theta}, \boldsymbol{\theta}_0) \lesssim \|T_{\boldsymbol{\theta}} - T_{\boldsymbol{\theta}_0}\|_{L^2(Q)}^2 + r_m^{-1}.
$$

Now we try to obtain the upper bound for $v_m(\boldsymbol{\theta}, \boldsymbol{\theta}_0)$. By triangle inequality,

$$
\begin{aligned}
&|d_\mathcal{W}(T_{\boldsymbol{\theta}}\#\hat{\mu}^m, \hat{\nu}^m) - d_\mathcal{W}(T_{\boldsymbol{\theta}_0}\#\hat{\mu}^m, \hat{\nu}^m)| \\
\leq\ &|d_\mathcal{W}(T_{\boldsymbol{\theta}}\#\hat{\mu}^m, \hat{\nu}^m) - d_\mathcal{W}(T_{\boldsymbol{\theta}}\#\mu, \nu)| + |d_\mathcal{W}(T_{\boldsymbol{\theta}}\#\mu, \nu) - d_\mathcal{W}(T_{\boldsymbol{\theta}_0}\#\mu, \nu)| \\
&+ |d_\mathcal{W}(T_{\boldsymbol{\theta}_0}\#\mu, \nu) - d_\mathcal{W}(T_{\boldsymbol{\theta}_0}\#\hat{\mu}^m, \hat{\nu}^m)|
\end{aligned}
$$

Similar calculations as above lead to

$$
|d_\mathcal{W}(T_{\boldsymbol{\theta}}\#\hat{\mu}^m, \hat{\nu}^m) - d_\mathcal{W}(T_{\boldsymbol{\theta}}\#\mu, \nu)| \lesssim r_m^{-1}
$$

and

$$
|d_\mathcal{W}(T_{\boldsymbol{\theta}_0}\#\mu, \nu) - d_\mathcal{W}(T_{\boldsymbol{\theta}_0}\#\hat{\mu}^m, \hat{\nu}^m)| \lesssim r_m^{-1}.
$$

We also have that

$$
\begin{aligned}
|d_\mathcal{W}(T_{\boldsymbol{\theta}}\#\mu, \nu) - d_\mathcal{W}(T_{\boldsymbol{\theta}_0}\#\mu, \nu)| &\leq d_\mathcal{W}(T_{\boldsymbol{\theta}}\#\mu, T_{\boldsymbol{\theta}_0}\#\mu) \\
&= \|T_{\boldsymbol{\theta}} - T_{\boldsymbol{\theta}_0}\|_{L^2(\mu)} \\
&\lesssim \|T_{\boldsymbol{\theta}} - T_{\boldsymbol{\theta}_0}\|_{L^2(Q)}.
\end{aligned}
$$

Since

$$
v_m(\boldsymbol{\theta}, \boldsymbol{\theta}_0) \lesssim P(|d_\mathcal{W}(T_{\boldsymbol{\theta}}\#\hat{\mu}^m, \hat{\nu}^m) - d_\mathcal{W}(T_{\boldsymbol{\theta}_0}\#\hat{\mu}^m, \hat{\nu}^m)|^2),
$$

it follows that

$$
v_m(\boldsymbol{\theta}, \boldsymbol{\theta}_0) \lesssim \|T_{\boldsymbol{\theta}} - T_{\boldsymbol{\theta}_0}\|_{L^2(Q)}^2 + r_m^{-2}.
$$

$\square$

We are now in a position to prove Theorem 2.

*Proof of Theorem 2.* Let $A_n := \{\boldsymbol{\theta} \in \Theta : ||T_{\boldsymbol{\theta}} - T_{\boldsymbol{\theta}_0}||_{L^2(Q)} > M\epsilon_n\}$. The Gibbs posterior probability of $A_n$ is given by

$$
\begin{aligned}
\Pi_n(A_n) &= \frac{N_n^m(A_n)}{D_n^m} \\
&= \frac{\int_{A_n} \exp\big(-\omega n\big(\tilde{R}_n^m(\boldsymbol{\theta}) - \tilde{R}_n^m(\boldsymbol{\theta}_0)\big)\big)\Pi(d\boldsymbol{\theta})}{\int_{\Theta} \exp\big(-\omega n\big(\tilde{R}_n^m(\boldsymbol{\theta}) - \tilde{R}_n^m(\boldsymbol{\theta}_0)\big)\big)\Pi(d\boldsymbol{\theta})}.
\end{aligned}
$$

Note that $m(n)$ is assumed to be a deterministic function of $n$. We aim to show that $P^n\Pi_n(A_n) \to 0$ as $n \to \infty$. We define the set

$$
G_n^m := \{\boldsymbol{\theta} \in \Theta : u_m(\boldsymbol{\theta}, \boldsymbol{\theta}_0) \leq \epsilon_n^2, v_m(\boldsymbol{\theta}, \boldsymbol{\theta}_0) \leq \epsilon_n^2\}.
$$

By Lemma 2 and the assumption $\epsilon_n^2 > r_m^{-1}$, $G_n^m$ is implied by the event

$$
H_n^m := \{\boldsymbol{\theta} \in \Theta : ||T_{\boldsymbol{\theta}} - T_{\boldsymbol{\theta}_0}||_{L^2(Q)}^2 \leq c(\epsilon_n^2 - r_m^{-1})\},
$$

for some constant $c > 0$. We thus have

$$
\Pi(G_n^m) \geq \Pi(H_n^m) \gtrsim \epsilon_n^{K+1},
$$

from which it follows that

$$
\log \Pi(G_n^m) \gtrsim -n\epsilon_n^2.
$$

Since the excess loss

$$
\ell_{\boldsymbol{\theta}}(\hat{\mu}^m, \hat{\nu}^m) - \ell_{\boldsymbol{\theta}_0}(\hat{\mu}^m, \hat{\nu}^m)
$$

is bounded for all $\boldsymbol{\theta}$ and $(\hat{\mu}^m, \hat{\nu}^m)$, when

$$
||T_{\boldsymbol{\theta}} - T_{\boldsymbol{\theta}_0}||_{L^2(Q)}^2 > \epsilon_n,
$$

we apply Section 3.4.1 of Syring and Martin [2023] to obtain

$$
\begin{aligned}
Pe^{-n\omega(\tilde{R}_n^m(\boldsymbol{\theta}) - \tilde{R}_n^m(\boldsymbol{\theta}_0))} &\leq \exp\Big(-nc_0\omega\Big(||T_{\boldsymbol{\theta}} - T_{\boldsymbol{\theta}_0}||_{L^2(Q)}^2 - r_{m(n)}^{-1} - r_{m(n)}^{-2}\Big)\Big) \\
&\leq \exp\big(-nc_1\omega(\epsilon_n^2 - r_m^{-1})\big) \\
&\leq \exp\big(-nc_2\omega\epsilon_n^2\big)
\end{aligned}
$$

for some constants $c_0, c_1, c_2 > 0$. By Fubini's Theorem, we have

$$
P^n N_n(A_n) = \int_{A_n} Pe^{-\omega n(\tilde{R}_n^m(\boldsymbol{\theta}) - \tilde{R}_n^m(\boldsymbol{\theta}_0))}\Pi(d\boldsymbol{\theta}) \leq \exp\Big(-nc_2\omega M^2\epsilon_n^2\Big).
$$

Following essentially the same lines as the proof of Lemma 1 of Syring and Martin [2023], we obtain

$$
P^n\left(D_n^m > \frac{1}{2}\Pi(G_n^m)e^{-2\omega n\epsilon_n^2}\right) \to 1,
$$

as $n \to \infty$.

Let $b_n^m = \frac{1}{2}\Pi(G_n^m)e^{-2\omega n\epsilon_n^2}$, we have

$$
P^n(D_n^m \leq b_n^m) \to 0
$$

as $n \to \infty$. Since

$$
\begin{aligned}
\Pi_n(A_n) &\leq \frac{N_n^m(A_n)}{D_n^m}1(D_n^m > b_n^m) + 1(D_n^m \leq b_n^m) \\
&\leq b_n^{-1}N_n^m(A_n) + 1(D_n^m \leq b_n^m)
\end{aligned}
$$

It follows that

$$
P^n\Pi_n(A_n) \to 0
$$

as $n \to \infty$. The proof is completed.

$\square$

# B   ADDITIONAL SIMULATION STUDIES

In order to evaluate the robustness of our proposed model and posterior sampling strategy, we conduct additional simulation studies aimed at investigating its behavior under mis-specification. We conduct simulation studies to investigate the behavior of the model under two scenarios of mis-specification. The first scenario involves the specified models having fewer basis functions than the true optimal transport maps. The second, more challenging scenario occurs when the true optimal transport maps do not belong to the model class.

In the first scenario, we replicate the simulation settings described in the main article, wherein the true optimal transport map is generated using BP basis functions with a polynomial order of $K = 50$. However, in the model fitting process, we set the polynomial order to $K = 20$. The outcomes of these experiments are illustrated in Figures 6, 7, and 8. Upon examination of the results, we observe that despite the mis-specification in the model fitting, the true optimal transport maps are successfully recovered in all scenarios. This suggests that our proposed model and posterior sampling strategy exhibit robustness to mis-specification, demonstrating their effectiveness in capturing underlying patterns even when the model assumptions are not entirely met.

In the second scenario, the true optimal transport maps cannot be expressed as linear combinations of BP basis functions. We conduct two simulations, and the results are shown in Figures 9 and 10. In the first case, the true optimal map can still be well approximated by BP basis functions, allowing the true optimal map to be well estimated. In the second scenario, the true optimal map is a step function with discontinuities and cannot be well approximated using BP basis functions. Consequently and not surprisingly, the estimated maps do not capture the shape of the true map.

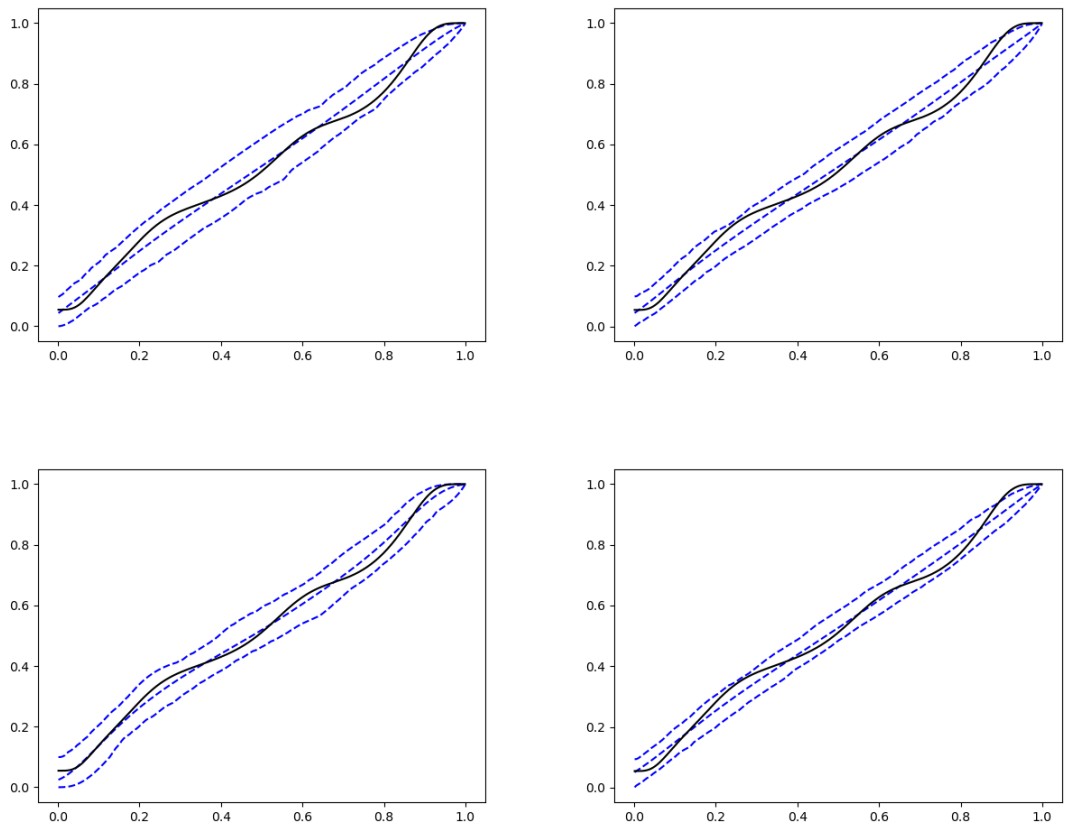

Figure 6: Simulation 1. Top left: n=5 ($\omega = 500$). Top right: n=20 ($\omega = 200$). Bottom left: n=50 ($\omega = 100$). Bottom right: n=100 ($\omega = 50$). Black curve: True optimal transport map. Blue dashed curves: estimated posterior mean and posterior credible intervals of optimal transport map.

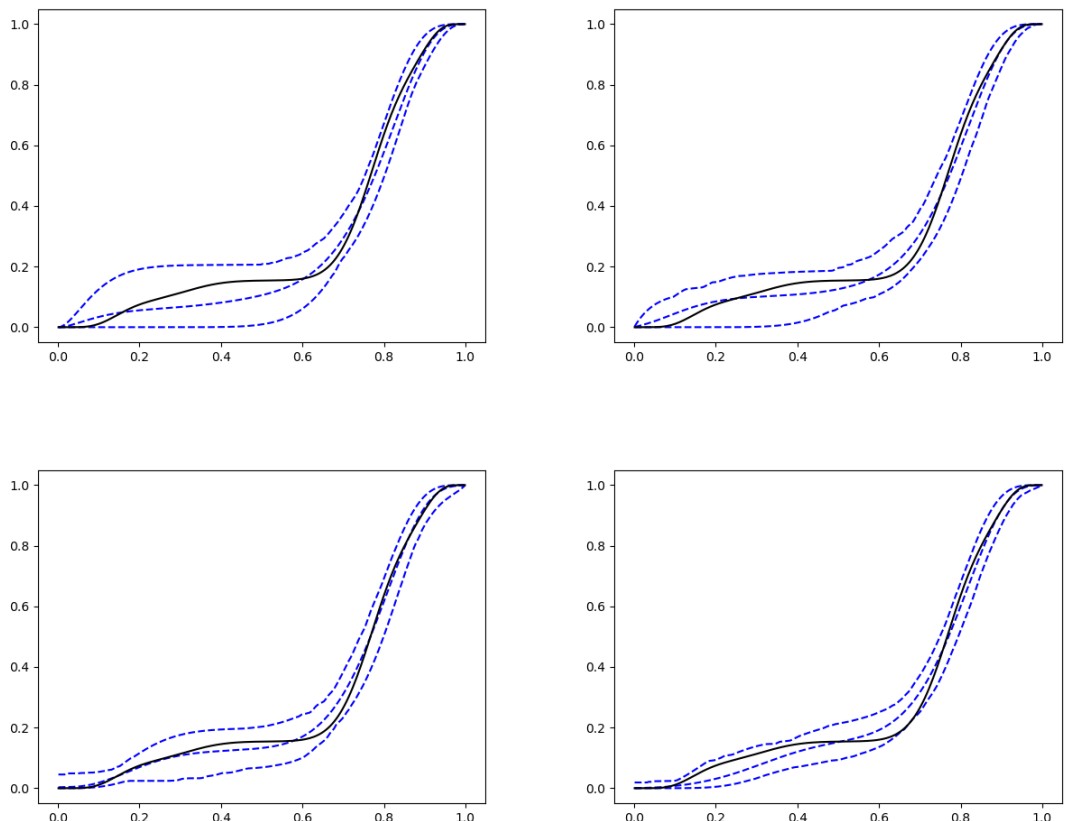

Figure 7: Simulation 2. Top left: n=5 ($\omega = 500$). Top right: n=20 ($\omega = 200$). Bottom left: n=50 ($\omega = 100$). Bottom right: n=100 ($\omega = 50$). Black curve: True optimal transport map. Blue dashed curves: estimated posterior mean and posterior credible intervals of optimal transport map.

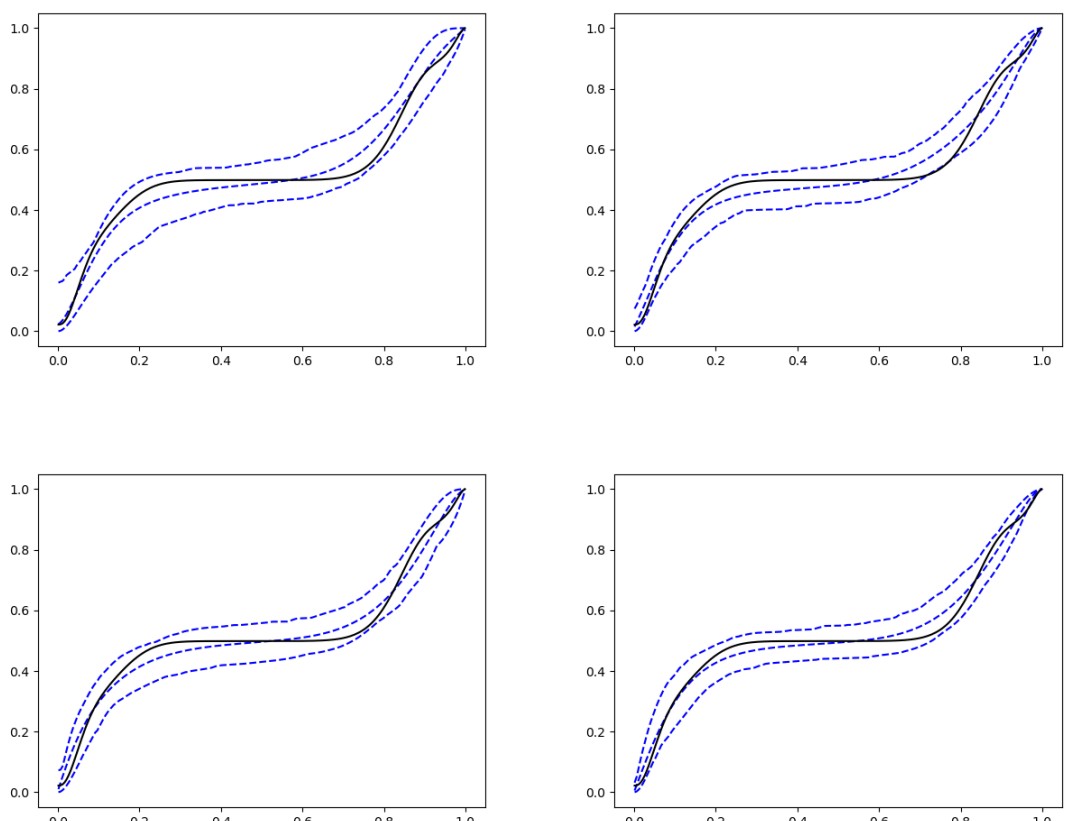

Figure 8: Simulation 3. Top left: n=5 ($\omega = 500$). Top right: n=20 ($\omega = 200$). Bottom left: n=50 ($\omega = 100$). Bottom right: n=100 ($\omega = 50$). Black curve: True optimal transport map. Blue dashed curves: estimated posterior mean and posterior credible intervals of optimal transport map.

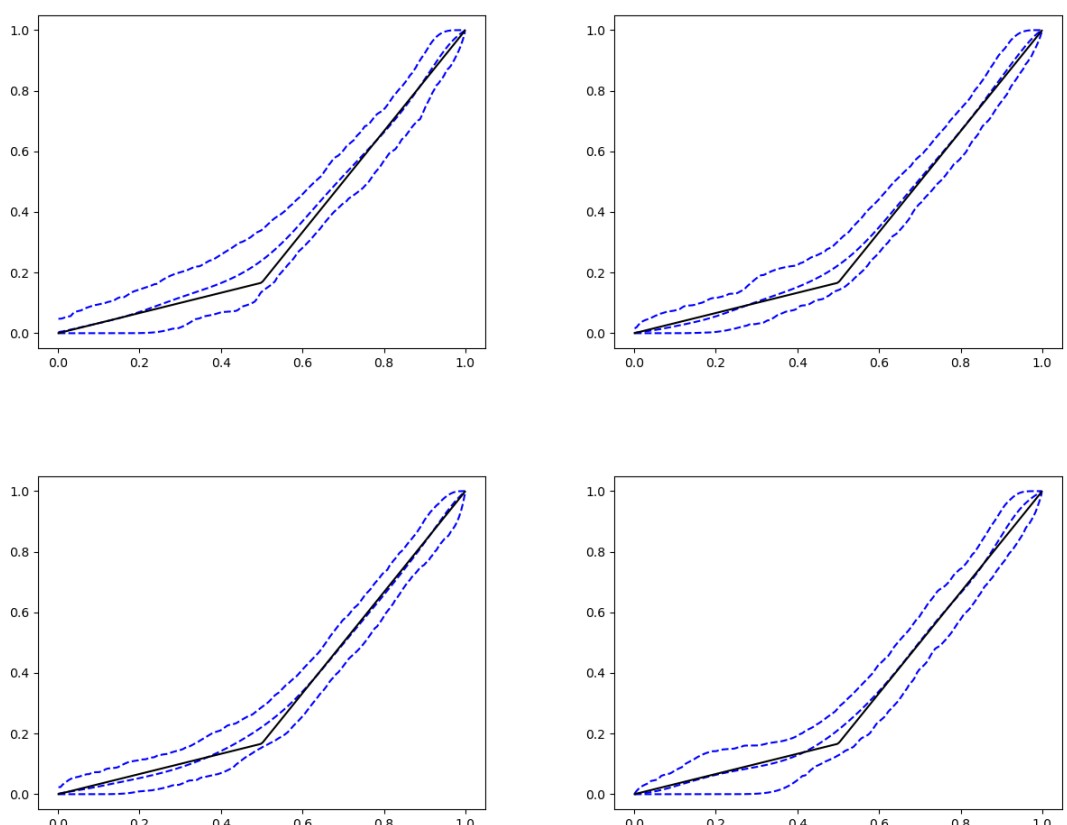

Figure 9: Simulation 4. Top left: n=5 ($\omega = 500$). Top right: n=20 ($\omega = 200$). Bottom left: n=50 ($\omega = 100$). Bottom right: n=100 ($\omega = 50$). Black curve: True optimal transport map. Blue dashed curves: estimated posterior mean and posterior credible intervals of optimal transport map.

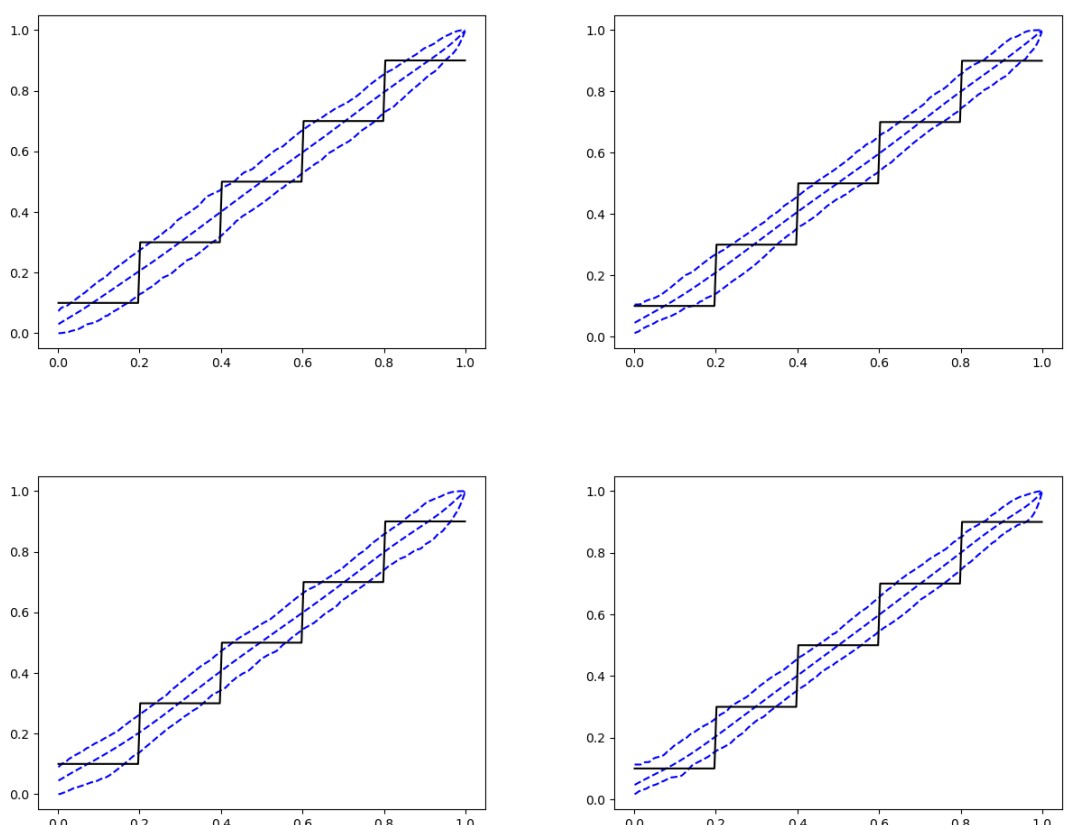

Figure 10: Simulation 5. Top left: n=5 ($\omega = 500$). Top right: n=20 ($\omega = 200$). Bottom left: n=50 ($\omega = 100$). Bottom right: n=100 ($\omega = 50$). Black curve: True optimal transport map. Blue dashed curves: estimated posterior mean and posterior credible intervals of optimal transport map.