# OpenReview forum: "A Generalized Bayesian Approach to Distribution-on-Distribution Regression"
_auai.org/UAI/2024/Conference — UAI 2024 poster_

### Official Review · Reviewer_CDby · 2024-03-04

**Q2-1 Originality-Novelty:** 2
**Q2-2 Correctness-Technical Quality:** 3
**Q2-5 Clarity Of Writing:** 3

**Q1 Summary And Contributions:**

The authors propose a generalized Bayesian approach for distribution-on-distribution regression. They provide the asymptotic analysis and MCMC algorithm. Simulation experiments and a real-data application further demonstrate their idea.

**Q2-3 Extent To Which Claims Are Supported By Evidence:**

2: Fair: the main claims are somewhat supported by evidence (but the experimental evaluation may be weak, or does not match entirely with the claims, important baselines may be missing, proofs contain important ideas but lack rigor, algorithmic details are only discussed superficially, references are imprecise, assumptions are not sufficiently motivated or explicated, etc.).

**Q2-4 Reproducibility:**

2: Fair: key resources (e.g. proofs, code, data) are unavailable but key details (e.g. proof sketches, experimental setup) are sufficiently well-described for an expert to confidently reproduce the main results.

**Q3 Main Strengths:**

The authors propose a generalized Bayesian approach for distribution-on-distribution regression with convergence guarantees. The message is clear.

**Q4 Main Weakness:**

Some assumptions need to be further justified, and comparisons with existing methods are lacking. See below for detailed comments.

**Q5 Detailed Comments To The Authors:**

*... providing a principled means to incorporate prior information and a formal framework for understanding and quantifying uncertainty ...* How do you incorporate prior information compared to the existing frequentist framework Ghodrati and Panaretos [2022], especially for the case study? For example, if we roughly know the shape of transport maps, how to incorporate that into your construction, and will this lead to better convergence or empirical performance? And why quantifying uncertainty is important for the problem of distribution-on-distribution regression? This point is not adequately addressed by your simulation experiments. Meanwhile, if the measures are not observed, there also should be uncertainty in estimating these measures from samples. This type of uncertainty seems to be ignored.

*... $\omega > 0$ is called the learning rate parameter ... we select $\omega$ as a function of the sample size ...* What function of the sample size is selected? Is it related to your theoretical analysis? This is not a very principled way.

In equation 7, should it be $P(\nu | \mu)$ instead of $P(\mu | \nu)$?

The authors choose to use Bernstein polynomials to approximate transport maps. Are there other choices and why BP is superior to others? They also assume the true regression operator is a transport map, which can be expressed using BP basis functions. The choice is ok, but I expect more rigorous justifications. For example, how well do BP basis functions approximate monotonic functions?

In theorem 1, is the rate minimax optimal (up to a logarithmic term)? Or how does it compare to the existing frequentist method? Such comparisons are also lacking in the simulation.

In theorem 2, for the case of imperfect observations, how to achieve the desired rate $r_m$, or if the empirical measures satisfy the condition?

For the choice of $K$, do you assume the truth in your theoretical development? It is more common to choose $K$ adaptively.

**Q9 Complying With Reviewing Instructions:**

Yes

---

> ### Author Rebuttal · Authors · 2024-04-04
>
> We thank the reviewer for the valuable comments and insightful questions. Here are some responses to your questions:
> \
> \
> Incorporation of prior information:
> \
> \
> Incorporation of prior information is relatively straightforward. Consider the parameterization of the optimal transport map:
> $$ T_{\boldsymbol{\theta}}(x) = \sum_{k=0}^{K} \theta_k G_{B(k,K-k+1)}(x) ,$$
> If we possess prior knowledge regarding the shape of the map, e.g. it increases sharply for small $x$ and more gradually for larger $x$, we can design the prior distribution on $\boldsymbol{\theta}$ accordingly. Specifically, we can structure the prior distribution so that $\theta_k$ has a higher probability of taking larger values for smaller values of $k$, while $\theta_k$ has a higher probability of taking smaller values for larger values of $k$. It is quite easy to construct prior distributions satisfying these conditions.
> \
> For example, in Section 3.2, instead of sampling $u_k$ as
> $$ u_k \sim \gamma_k \mbox{Unif}(0,1) + (1-\gamma_k) \delta_{0} ,$$
> we can have
> $$ u_k \sim \gamma_k \mbox{Be}(a_k, b_k) + (1-\gamma_k) \delta_{0},$$
> for suitable choices of $a_k, b_k$ and where $\mbox{Be}(\cdot,\cdot)$ is the Beta distribution.
> \
> \
> Adopting a prior such as above will not affect the contraction rate. As explained in the paper, this is due to the fact that in the finite dimensional case, it is impossible for a fixed prior to assign mass bounded away from $0$ to a shrinking neighborhood of $\boldsymbol{\theta}_0$.
> \
> \
> Empirically, the impact of the choice of prior on the resulting Gibbs posterior largely hinges on whether the data aligns with or contradicts the prior. While there exists a vast body of literature on this topic, we believe that delving into this aspect is beyond the scope of our paper.
> \
> \
> \
> \
> Importance of uncertainty quantification for distribution-on-distribution regression:
> \
> \
> We would argue that uncertainty quantification is important and valuable in any statistical estimation problem. In the context of distribution-on-distribution regression, it provides information on the accuracy of the estimated map.
> \
> \
> \
> \
> Uncertainty in estimating measures from samples:
> \
> \
> The reviewer is correct that the uncertainty associated with estimating the measures from samples is not incorporated in the model. However, this is a deliberate modeling choice within the Gibbs posterior framework. This framework enables us to concentrate solely on specific aspects of the data generating process, the optimal transport map in our case. This differs from the standard Bayesian framework, where the entire data generating process must be fully specified. As mentioned in the paper, one of the key advantages of the Gibbs posterior framework over the standard Bayesian approach is that it mitigates the need for fully specifying the entire data generating process, which can be cumbersome and increase the risk of model mis-specification.
> \
> \
> We will emphasize this in the paper.
> \
> \
> \
> \
> The choice of the learning rate:
> \
> \
> Please see our response to Reviewer XEjm.
> \
> \
> \
> \
> Equation (7):
> \
> \
> The reviewer is correct that it should be $P(\nu|\mu)$. Thank you for spotting the typo.
> \
> \
> \
> \
> Choice of Bernstein polynomials:
> \
> \
> Please refer to our response to reviewer L7n5 concerning the justification of the BP basis. We acknowledge that alternative choices, such as transformed spline basis, are feasible. Moreover, it's worth noting that analogous contraction rates can be derived since our proof technique does not rely on properties specific to the BP basis. Alternative nonparametric approaches such as monotone Gaussian process can also be considered. We will add a short discussion to the paper.
> \
> \
> \
> \
> Convergence rate in Theorem 1:
> \
> \
> The frequentist minimax rate is at least $n^{-1/3}$ from (Ghodrati & Panaretos 2022). Our posterior contraction rate of $n^{-1/2} (\log n)^{1/2}$ is faster. However, this is not an appropriate comparison since we are operating under the parametric setting.
> \
> \
> Numerical comparison (e.g. convergence behavior) is not straightforward and can be misleading due to the inherent differences between our Bayesian approach and the frequentist setting. In the frequentist setting, only a point estimate is available, whereas our focus lies on posterior coverage and contraction.
> \
> \
> \
> \
> Achieve the desired rate $r_m$:
> \
> \
> In general, the empirical measure does not achieve the desired rate (Theorem 1 and Proposition 7 of Weed & Bach, Bernoulli 2019),
> it has a slightly slower convergence rate than required in Theorem 2. However, for measures possess smooth densities, the required rate can be achieved (Niles-Weed & Berthet, Annals of Statistics 2022).
> \
> \
> \
> \
> Choice of $K$:
> \
> \
> The reviewer is correct that we assume the true value of $K$ in the theoretical results. While one could for example assign a prior on $K$ or let $K = K_n$ be a function of sample size $n$, the resulting theoretical analysis would become substantially more complicated.

---

### Official Review · Reviewer_XYKf · 2024-03-20

**Q2-1 Originality-Novelty:** 3
**Q2-2 Correctness-Technical Quality:** 3
**Q2-5 Clarity Of Writing:** 3

**Q1 Summary And Contributions:**

The authors propose the first generalized Bayesian approach for distribution-on-distribution regression by employing the 2-Wasserstein distance. In their parametric approach, they define transport maps from a covariate distribution to a response distribution using Bernstein Polynomial basis functions. Under the assumption that the optimal transport map conforms to this form, they show that the generalized (Gibbs) posterior distribution they defined contracts at rate $\sqrt{\frac{\log n}{n}}$ to the true optimal transport map parameterized by the unique minimizer of the expected risk. Finally, they propose a Gibbs sampling scheme for the posterior and show that the posterior mean and credible intervals adequately cover the true optimal transport map in both simulated and real-world experiments.

**Q2-3 Extent To Which Claims Are Supported By Evidence:**

3: Good: the main claims are supported by convincing evidence (in the form of adequate experimental evaluation, proofs, (pseudo-)code, references, assumptions).

**Q2-4 Reproducibility:**

3: Good: key resources (e.g. proofs, code, data) are available and key details (e.g. proofs, experimental setup) are sufficiently well-described for competent researchers to confidently reproduce the main results.

**Q3 Main Strengths:**

This work is preeminently methodological and seems to be of high technical quality. The approach proposed advanced the current state-of-the-art of distribution-on-distribution regression by introducing a first generalized Bayesian approach for solving this task. The claims are supported adequately, the paper is well written, and reasonably reproducible.

**Q4 Main Weakness:**

The paper has no significant weaknesses, although there are certain details lacking clarity, as described in Q5.

**Q5 Detailed Comments To The Authors:**

- In Section 2.3, why is the target monotone function "relatively flat in some sub-interval $(a, b) \subset [0,1]$?
- In Section 3.1 you highlight the main assumption that $\boldsymbol{\theta_0} \in \Theta$, yet in the conclusion you focus on the main assumption that $T_{\theta_0}$ can be written as the sum of Bernstein polynomials basis functions. Aren't these two different assumptions? Could you elaborate on the assumptions made?
- In Section 3.4, what is $q_\gamma$? Don't you mean $p_\gamma$, the sparsity parameter? Both $p_\gamma$ and $q_\gamma$ should be explained more clearly before describing the Bayesian computation. Moreover, isn't this probability updated every step, so we should probably have $p_\gamma^{(t)}$ in the formula for $\tilde{\gamma_k}^{(t+1)}$, right?
- Is the only difference between the three simulation studies in Section 4 that the optimal transport map "$T_{\theta_0}$"? This should be mentioned more explicitly. Is there any substantial difference between the optimal transport maps such that three simulations need to be shown?
- For Figure 4, is the *y*-axis correct? Shouldn't it correspond to the target distribution, so the range should be the similar to that of the *x*-axis? Legends and captions could be more detailed in general.

Writing comments:
- Page 2, left column, Section 2.1: "Borel **sets**"
- Page 2, right column, last paragraph: "asymptotically **concentrate**"; "arbitrarily **slowly**"; "$P^n f$ **denotes**"; "The **general** conditions..."
- Page 3, Section 2.4: "$a \le Cb$ **holds**"
- Page 3, Section 3.1: It is slightly confusing to have both $\theta_0$ and $\boldsymbol{\theta}_0$ representing different parameters.
- Page 3, right column, paragraph after (7): "left unspecified and **assumed** to be a monotone"
- Page 5, right column, after (11): "In order for the Gibbs posterior distribution **to contract** around..."; I think you mean "a deterministic function of **n**"
- Page 6, left column, first full paragraph: "**with** probability = $q_\gamma$"
- Page 6, left column, last paragraph before Section 4: "setting it **to** the order"
- Page 6, right column, last full paragraph: "map" is duplicated before "$T_{\theta_0}$"
- References: it might be good to specify that some references come from *arXiv*.

**Q9 Complying With Reviewing Instructions:**

Yes

---

> ### Author Rebuttal · Authors · 2024-04-04
>
> We thank the reviewer for the valuable comments and insightful questions. Here are some responses to your questions:
> \
> \
> \
> \
> Target monotone function "relatively flat in some sub-interval" $(a,b) \subset [0,1]$:
> \
> \
> The sentence in Section 2.3 reads "This re-parameterization allows modeling the target monotone function that is relatively flat in some sub-interval $(a,b) \subset [0,1]$ ". We intended to convey that re-parameterization is advantageous in scenarios where the target monotone function exhibits relative flatness within certain sub-intervals. We will re-word this sentence in the paper to avoid confusion.
> \
> \
> \
> \
> Assumption concerning $\boldsymbol{\theta_0}$:
> \
> \
> These two statements are equivalent. Having $\boldsymbol{\theta_0} \in \Theta$ is equivalent to having the true optimal transport map taking the form
> $$ T_{\boldsymbol{\theta_0}}(x) = \sum_{k=0}^{K} \theta_k G_{B(k,K-k+1)}(x) ,$$
> for some $\theta_0, \theta_1, \ldots, \theta_K > 0$ with $\sum_{k=0}^{K} \theta_k = 1$. This is equivalent to requiring the true map takes the form of convex combination of the basis functions $ G_{B(k,K-k+1)}(\cdot), k=0, \ldots, K$.
> \
> \
> \
> \
> Difference between $q_{\gamma}$ and $p_{\gamma}$:
> \
> \
> $q_{\gamma}$ and $p_{\gamma}$ are different. While $p_{\gamma}$ is the prior parameter controlling the sparsity of the binary variables $\gamma_k, k=0,1,\ldots,K$, $q_{\gamma}$ is a hyper-parameter for updating $\gamma_k,k=0,1,\ldots,K$ in the MCMC. In particular, $1-q_{\gamma}$ represents the probability that the proposed $\gamma_k$ at iteration $t+1$ equals the current value of $\gamma_k$ at iteration $t$
> \
> \
> When updating $\gamma_k$, the sparsity parameter $p_{\gamma}$ appears in the Metropolis acceptance ratio.
> \
> \
> We will clarify these in the paper.
> \
> \
> \
> \
> Difference between the three simulation studies:
> \
> \
> The reviewer is correct in noting that the three simulation studies employ three different optimal maps, each representing distinct scenarios. In the first case, the true map closely resembles a straight line. In the second case, the map exhibits slow increase in the sub-interval $(0,0.7)$ and rapid increase in $(0.7,1)$. In the third case, the true map is relatively flat in $(0.2,0.7)$. We will consider relocating one of the simulation settings to the supplement to allow for a more detailed description of the algorithm.
> \
> \
> \
> \
> y-axis in Figure 4:
> \
> \
> Thank you for spotting the error in the y-axis of Figure 4. We will correct this.
> \
> \
> \
> \
> Thank you for your writing comments. We will revise the paper accordingly.

---

### Official Review · Reviewer_XEjm · 2024-03-22

**Q2-1 Originality-Novelty:** 3
**Q2-2 Correctness-Technical Quality:** 3
**Q2-5 Clarity Of Writing:** 3

**Q1 Summary And Contributions:**

The problem of distribution-on-distribution regression is discussed (the covariate is a distribution, and a regression function must learn to map it to a new distribution). A generalized Bayesian approach is developed here for this problem, some its properties are shown theoretically, and it is empirically validated on simulated and real-world data.

**Q2-3 Extent To Which Claims Are Supported By Evidence:**

3: Good: the main claims are supported by convincing evidence (in the form of adequate experimental evaluation, proofs, (pseudo-)code, references, assumptions).

**Q2-4 Reproducibility:**

3: Good: key resources (e.g. proofs, code, data) are available and key details (e.g. proofs, experimental setup) are sufficiently well-described for competent researchers to confidently reproduce the main results.

**Q3 Main Strengths:**

I wasn't familiar with distribution-on-distribution regression, but can see it has applications, and agree that a Bayesian method for this problem offers important benefits over existing methods. The paper managed to describe the material clearly; the contribution is quite technical, but looks to be of solid technical quality. (I was not able to verify everything; in particular I did not check the proofs in the supplement.)

**Q4 Main Weakness:**

None of my concerns below are major.

**Q5 Detailed Comments To The Authors:**

EDIT: I'd like to thank the authors for their clarifications in the rebuttal.

Questions / comments:
- Is the assumption in section 3.1 that $\Omega = [0,1]$ truly without loss of generality? Nonlinear transformations would deform the Wasserstein metric, and with only linear transformations, an unbounded set can't be transformed to $[0,1]$.
- Could you explain in the paper in a bit more detail what is happening in equation (7)? I could not fully match the parts of the equation to the textual description.
- How were the values of the learning rate chosen - did you choose these experimentally? Please include in this section the details of how you approached this.
- Possibly related to the above, the results in Figure 1 look oversmoothed: the posteriors look roughly straight, and don't follow the bends of the true optimal transport map. Is this expected in this setting? Or could the chosen learning rates be suboptimal here?

Typographical and other minor comments:
- PCA is princip*al* component analysis
- end of S1: "the" should be "a data application"
- it's odd that equation (9) is referenced a few times well before it appears
- above (4): "concentrates" -> "concentrate"
- below (4): "General" -> "general"
- below (7): "to standard" -> "to the standard"
- next paragraph: "assume to be" -> "assumed to be"
- equation (9) ends with a comma, should be a period
- final sentence of section 3.2 is ungrammatical (was a word left out?)
- first display in section 3.4: "wtih" -> "with"
- halfway section 4: "We then simulate $J-1$ uniform random variable" - add "s"
- a bit lower: double "map"
- include a reference to Appendix B in the main text
- References: "Zhenhua Lin Yaqing Chen" should be two names; the final reference has dollar signs in the title

**Q9 Complying With Reviewing Instructions:**

Yes

---

> ### Author Rebuttal · Authors · 2024-04-04
>
> We thank the reviewer for the valuable comments and insightful questions. Here are some responses to your questions:
> \
> \
> Assumption of $\Omega$:
> \
> As mentioned in the Background section (Section 2.1), we are considering the setting that $\Omega$ is a compact set. We will emphasize this in the beginning of Section 3.1.
> \
> \
> \
> \
> Explanation of Equation (7):
> \
> \
> Equation (7) provides the definition of the regression operator $\Gamma$. We can see how this is connected with the ``standard'' regression setting as follows:
> \
> \
> Recall that in the standard regression setting with covariate $X \in \mathbb{R}^d$ and scalar response $Y \in \mathbb{R}$, we can define the regression operator $f: \mathbb{R}^d \rightarrow \mathbb{R}$ as follows:
> \
> \
> $$ f(x) := \mbox{argmin}_{w \in \mathbb{R}} E(|w - Y|^2|X=x) ,$$
> that is, for each fixed $x \in \mathbb{R}^d$, $f(x)$ is defined as the value $w \in \mathbb{R}$ which minimizes the conditional expectation above. Equation (7) is analogous to the equality above, but instead of having a conditional expectation, we have the Wasserstein-Fr\'echet mean.
> \
> \
> We will add further explanation in the paper.
> \
> \
> \
> \
> Choice of $\omega$:
> \
> \
> The choice of the learning rate $\omega$ is indeed a challenging and open problem (Wu & Martin, Bayesian Analysis 2023). We adopt the strategies of (Syring & Martin, Biometrika 2019) and (Syring & Martin, Bernoulli 2023) to let $\omega$ be a decreasing function of the sample size $n$ and also tune $\omega$ to ensure the resulting Gibbs posterior has sufficient coverage probability. We will add further explanation regarding this approach in the paper.
> \
> \
> \
> \
> Figure 1:
> \
> \
> The true optimal transport map resembles a straight line with some wiggling. The smooth behavior of the Gibbs posterior is partly explained by the choice of the Bernstein polynomial basis functions (which are smooth) and that the "signal" present in the true map is quite small relative to the "noise" induced by the random error maps. We will add a short explanation on this in the simulation section.
> \
> \
> \
> \
> Thank you for spotting the typographical errors. We will correct them.

---

### Official Review · Reviewer_L7n5 · 2024-03-27

**Q2-1 Originality-Novelty:** 3
**Q2-2 Correctness-Technical Quality:** 4
**Q2-5 Clarity Of Writing:** 4

**Q1 Summary And Contributions:**

This work studies the problem of distribution on distribution regression, where both the input and outputs are in form of distributions. The authors define the regression operator in terms of the minimization of a Frechet functional and model the mapping using a parametric approach by employing basis functions based on Bernstein polynomials, under the assumption that the problem is well specified, i.e. the true parameter lies within the span of this basis. The authors then define a prior on the coefficients, based on the approach employed in Curtis and Ghosh (2011), and prove concentration in both the case where the distributions are specified exactly and the case where they must be estimated from samples (the "noisy" case). A sampling procedure is then described for estimating parameters in practice. Simulation studies show the performance of the proposed estimators on different functions, and an application is provided using mortality data.

**Q2-3 Extent To Which Claims Are Supported By Evidence:**

2: Fair: the main claims are somewhat supported by evidence (but the experimental evaluation may be weak, or does not match entirely with the claims, important baselines may be missing, proofs contain important ideas but lack rigor, algorithmic details are only discussed superficially, references are imprecise, assumptions are not sufficiently motivated or explicated, etc.).

**Q2-4 Reproducibility:**

3: Good: key resources (e.g. proofs, code, data) are available and key details (e.g. proofs, experimental setup) are sufficiently well-described for competent researchers to confidently reproduce the main results.

**Q3 Main Strengths:**

The authors tackle a practically relevant and challenging problem and provide an elegant solution. The use of a basis using Bernstein polynomials is both natural and allows for a relatively simple approach to modeling, the parametric approach. The authors do a nice job of motivating the choices at each step. The results on the concentration of the proposed approach are particularly nice (though under strong assumptions as I talk about below), and empirical results show the proposed method performing sensibly.

**Q4 Main Weakness:**

* The grounding of this method relies on the truth lying within the proposed parameter space. This is a fairly strong assumption and there isn't an analysis of when this will fail to hold
* There is very limited comparison to existing non-Bayesian methods and there are citations missing to some related work (e.g., Law et al. from AISTATS 2018 that studies a related problem of distribution on real regression).
* The experimental results are underwhelming and while they confirm that the algorithm works in some relatively simple settings they do little to give meaningful understanding of the behavior of the proposed approach.

**Q5 Detailed Comments To The Authors:**

As I mentioned above, overall I think this work is very interesting and well motivated. My main concern here is that given that there is a strong assumption made in order to establish properties of the proposed work there needs to be an accompanying set of empirical results that examine the properties of the approach in different settings. For example, I would like to see settings where there is explicit misspecification. It would also be helpful to examine both running time and performance as a function of number of available examples, convergence of "noisy samples" by varying the number available. It's also not clear to me why there isn't any comparison to existing non-Bayesian methods. While I entirely understand that the proposed work stands on different footing in interpretation and theory there is no reason why there can't be _some_ comparison empirically to the frequentist methods mentioned in the introduction.

**Q9 Complying With Reviewing Instructions:**

Yes

---

> ### Author Rebuttal · Authors · 2024-04-04
>
> We thank the reviewer for the valuable comments and insightful questions. Here are some responses to the weaknesses and questions:
> \
> \
> Weakness 1:
> \
> Assuming that the ground truth optimal transport map lies within the parameter space is indeed a fairly strong assumption. This assumption is needed for bounding the excess expected risk $R(\boldsymbol{\theta}) - R(\boldsymbol{\theta}_0)$ in the proof of the main theorems.
> \
> \
> The rationale behind selecting Bernstein polynomial basis functions and assuming the truth lies within the parameter space is primarily motivated by the literature on monotone regression. Although the underlying problem of monotone regression differs substantially from ours, both involve parameterizing a monotone function. Empirical evidence from the monotone regression literature has validated that Bernstein polynomial basis functions offer the flexibility needed to model monotonic functions effectively. Furthermore, it is well known that any continuous function can be uniformly approximated by Bernstein polynomial basis functions, as per Weierstrass theorem. Consequently, with a large enough $K$, one would anticipate that the ground truth could be accurately approximated by elements within the parameter space.
> \
> \
> We would like to emphasize that our proof of the main results does not hinge on the properties of Bernstein polynomials. Consequently, analogous contraction rates can be derived for alternative choices of basis functions, such as transformed B-spline basis functions. We will add a short discussion on this in the paper.
> \
> \
> Weakness 2:
> \
> Regarding the comparison with frequentist approaches, as elaborated in Section 5, our data application using the mortality dataset shares the same setup as Ghodrati & Panaretos (2022) and Chen et al. (2023). This involves selecting the same countries and opting for the year 1983 as the covariate and the year 2013 as the response. Our main findings align consistently with those of Ghodrati & Panaretos (2022). However, comparing with the approach in Chen et al. (2023) poses more challenges, primarily because their regression operator is defined on the tangent space.
> \
> \
> Numerical comparison (e.g. convergence behavior) is not straightforward and can be misleading due to the inherent differences between our Bayesian approach and the frequentist setting. In the frequentist setting, only a point estimate is available, whereas our focus lies on posterior coverage and contraction.
> \
> \
> Thank you for your suggestion to include the reference from Law et al. (AISTATS 2018). We are aware that there are numerous related regression problems, including those involving real-valued response paired with distribution predictors (Law et al., 2018 and other works), Euclidean predictors paired with responses in Riemannian manifold, and scenarios where both responses and predictors lie in Riemannian manifold. We did not initially include these references because the distribution-on-distribution regression problem differs significantly, which typically involves working with Wasserstein space and optimal transport theory.
>
> We will incorporate a paragraph in the literature review section discussing some of these related works.
>
>
>
> \
> \
> Weakness 3:
> \
> Our simulation studies explore the performance of the method under various scenarios where the true optimal transport map exhibits significantly different shapes, including where the true map closely resembles a straight line, the true map exhibits slow increase in the interval $(0, 0.7)$ and sharp increase in $(0.7, 1)$, and the true map is nearly flat in the interval $(0.2,0.7)$. In the supplement, we extend our investigation to scenarios where the true map is more complex than the parameter space.
> \
> \
> Questions:
> \
> \
> Regarding additional experiments with explicit model mis-specification, we suspect that the method's performance will heavily depend on how well the Bernstein polynomial basis can approximate the truth. If the truth can be accurately approximated, we would anticipate the method performing well, akin to the results shown in the supplementary materials.
> \
> \
> We will include further simulation studies where the true optimal transport maps do not belong to the parameter space for any value of $K$.
> \
> \
> The computational complexity of the MCMC is linear with respect to both the sample size and the number of basis functions.
> \
> \
> Regarding performance in relation to sample size, in the perfectly observed scenario, as demonstrated in the simulation studies, the method performs well even with a sample size as small as $n=5$. This is expected since each distribution can be considered as containing an infinite amount of data.
> \
> \
> Exploring the method's behavior with varying numbers of "noisy samples" is more intricate, given the diverse choices of underlying distribution to generate the samples and that various methods can be used to estimate the CDFs based on the noisy samples.

---

### Meta-Review · Area_Chair_P8v2 · 2024-04-15

Overall, the paper provides a contribution to the growing literature on generalized Bayesian inference, here in the case of distribution-on-distribution regression. There was an overall consensus that this is a worthwhile addition to the literature, but the details raised by the reviewers should be addressed in a possible final version.